# Microstructural asymmetry in the human cortex

Bin Wan [1,2,3,4] ✉, Amin Saberi [1,4,5], Casey Paquola[4], H. Lina Schaare [1,4], Meike D. Hettwer [1,4,5,6], Jessica Royer[7], Alexandra John [1,4], Lena Dorfschmidt [8,9], Şeyma Bayrak[1,3,4], Richard A. I. Bethlehem[10], Simon B. Eickhoff[4,5], Boris C. Bernhardt[7] & Sofie L. Valk [1,4,5] ✉

The human cerebral cortex shows hemispheric asymmetry, yet the microstructural basis of this asymmetry remains incompletely understood. Here, we probe layer-specific microstructural asymmetry using one *post-mortem* male brain. Overall, anterior and posterior regions show leftward and rightward asymmetry respectively, but this pattern varies across cortical layers. A similar anterior-posterior pattern is observed using in vivo Human Connectome Project ($N = 1101$) T1w/T2w microstructural data, with average cortical asymmetry showing the strongest similarity with *post-mortem*-based asymmetry of layer III. Moreover, microstructural asymmetry is found to be heritable, varies as a function of age and sex, and corresponds to intrinsic functional asymmetry. We also observe a differential association of language and markers of mental health with microstructural asymmetry patterns at the individual level, illustrating a functional divergence between inferior-superior and anterior-posterior microstructural axes, possibly anchored in development. Last, we could show concordant evidence with alternative in vivo microstructural measures: magnetization transfer ($N = 286$) and quantitative T1 ($N = 50$). Together, our study highlights microstructural asymmetry in the human cortex and its functional and behavioral relevance.

Hemispheric specialization is a crucial aspect of brain organization that supports human cognitive functions, including language and attention[1–6]. Previous research has shown neuroanatomical differences between the left and right hemispheres at the macroscale[7–13]. Specifically, cortical thickness exhibits an asymmetrical pattern that extends from anterior (leftward asymmetry) to posterior (rightward asymmetry) regions[10,14]. These structural differences between the left and right hemispheres of the brain may have underlying genetic components, as suggested by Sha et al[12]. While macrostructural asymmetry has been widely studied, underlying micro and mesostructure markers such as cytoarchitecture and myeloarchitecture of cortical regions have largely been examined in regional isolation.

The neocortex is composed of six layers that contain neurons of varying size and density. These layers, arranged from the pial to gray-

[1]Otto Hahn Research Group Cognitive Neurogenetics, Max Planck Institute for Human Cognitive and Brain Sciences, Leipzig, Germany. [2]International Max Planck Research School on Neuroscience of Communication: Function, Structure, and Plasticity (IMPRS NeuroCom), Leipzig, Germany. [3]Department of Cognitive Neurology, University Hospital Leipzig and Faculty of Medicine, University of Leipzig, Leipzig, Germany. [4]Institute of Neuroscience and Medicine (INM-7: Brain and Behavior), Research Center Jülich, Jülich, Germany. [5]Institute of Systems Neuroscience, Medical Faculty and University Hospital Düsseldorfpital Düsseldorf, Heinrich Heine University Düsseldorf, Düsseldorf, Germany. [6]Max Planck School of Cognition, Leipzig, Germany. [7]McConnell Brain Imaging Centre, Montréal Neurological Institute and Hospital, McGill University, Montréal, QC, Canada. [8]Department of Child and Adolescent Psychiatry and Behavioral Science, The Children's Hospital of Philadelphia, Philadelphia, PA, USA. [9]Lifespan Brain Institute, The Children's Hospital of Philadelphia and Penn Medicine, Philadelphia, PA, USA. [10]Department of Psychology, University of Cambridge, Cambridge, UK. ✉e-mail: binwan@cbs.mpg.de; valk@cbs.mpg.de

white matter boundary, consist of layer I, which is rich in apical dendrites and axon terminals, layers II and III, rich in pyramidal cells, layer IV, with densely packed neurons, layer V, containing small (5a) or large (5b) pyramidal neurons, and layer VI, featuring corticothalamic pyramidal cells[15–17]. It is important to note that laminar and cytoarchitectonic features, which are crucial in qualitative studies, vary across the cortex. For instance, while sensory regions exhibit a well-laminated structure and high cell density, association areas, including language networks, display a reduced cell density and less distinct laminar structure[15,18–20]. There are a few studies focusing on microstructural asymmetry in language regions[7,21–24] and amygdala[25]. For example, microstructural intensity in language areas (BA areas 44 and 45) is higher in the left hemisphere[7] and left-right differences in amygdala subnuclear volumes measured by cytoarchitectural mapping[23]. Although evidence of asymmetry in cortical cytoarchitecture is limited, understanding this phenomenon could be crucial for advancing our knowledge of brain function.

Although *post-mortem* data can provide new insights of cortical microstructure and its potential asymmetry at the microscale, it cannot be linked directly to individual differences and potential functional relevance. Recent advancements in quantitative magnetic resonance imaging (MRI) have made it possible to obtain detailed region-wise in vivo microstructure information based on imaging markers such as T1w/T2w[8,26], quantitative T1 (qT1) relaxometry[27–30], and magnetization transfer (MT)[31–33]. In vivo quantitative MRI captures the higher intensity in sensory areas and lower intensity in transmodal dys- and agranular cortical regions[20,34]. This differentiation between sensory and transmodal regions is also present in intrinsic functional organization[35], suggesting a common principle of brain organization for microstructure and function[34]. Such a principle would be in line with the structural model, posing that regions with similar microstructure may be functionally connected[16]. Indeed, various studies have demonstrated asymmetry along this sensory-transmodal-axis for intrinsic function[6,36–39].

Motivated by previous work showing cortex-wide patterns of asymmetry in macrostructural markers such as cortical thickness and surface area[10,12,14], and regional reports of asymmetry in cortical microstructure[7,21–25], we aimed to study microstructural asymmetry across the whole cortex. Specifically, we probed the microstructural basis of cortical asymmetry using a multiscale approach based on high-resolution histology and imaging data. Given that studies on macro-scale structural asymmetry have reported leftward asymmetry in frontal regions and rightward asymmetry in occipital regions, we wished to study what the underlying microstructural correlates of these macrostructural patterns would be. First, we studied the BigBrain, which is an ultra-high-resolution whole-brain *post-mortem* histological atlas of a 65-year-old male. It allowed the quantification of asymmetry in cortical cytoarchitecture at the level of individual layers[15,40–42]. Second, we studied in vivo microstructural asymmetry to evaluate inter-individual variation. For in vivo microstructural maps we used the T1w/T2w ratio from 1101 individual images from the Human Connectome Project (HCP) from young adults[26,43]. We furthermore aimed to probe its functional relevance, motivated by the Structural Model, stating that microstructural similarity relates to connectivity[16], and previous work on the functional markers of asymmetry[6,13,14]. Structural brain asymmetry is linked to behavioral differences such as variability in language skills[44–46] and mental health[47] such as autism[48,49], attention-deficit/hyperactivity disorder[50], schizophrenia[51], and substance dependence[52]. Finally, based on the healthy HCP sample, we investigated the associations between microstructural asymmetry and individual differences in language skills, as well as its potential relevance to mental health traits, including depression, anxiety, somatic, avoidant, ADHD, and antisocial phenotypes. Given that different imaging sequences have been proposed to measure microstructure in vivo, as mentioned above, we leveraged these measures to verify our results, including qT1

relaxometry from a dataset ($N = 50$) for microstructure-informed connectomics (MICs) in young adults[28] and MT maps from a longitudinal cohort of adolescents and young adults ($N = 286$) acquired as part of the Neuroscience In Psychiatry Network (NSPN).

## Results

### Differentiable patterns of microstructural asymmetry as a function of cortical depth in a ultra-high resolution *post-mortem* sample (Fig. 1)

We first mapped the cortical cytoarchitecture asymmetry using ultra-high resolution *post-mortem* data based on the BigBrain[15,53]. The sliced sections (20 μm) of the BigBrain were cell-body stained, scanned and reconstructed in 3D, resulting in an ultra-high resolution atlas (100 μm³) (Fig. 1a). Using the cortical cell-staining intensity of BigBrain as a feature, a six-layer cortical segmentation (60 surfaces) was obtained of the whole cerebral cortex via a convolutional neural network algorithm[42]. Multimodal[54] and Cole-Anticevic (CA) parcellation[55] were utilized to downsample the maps into 360 regions and 12 networks. The CA networks consisted of primary visual (Vis1), secondary visual (Vis2), somatomotor (SMN), cingulo-opercular (CON), dorsal attention (DAN), language (LAN), frontoparietal (FPN), auditory network (AUD), default mode (DMN), posterior multimodal (PMN), ventral multimodal (VMN), and orbito-affective (OAN). To prevent measurement bias, the mean intensity for layer profiles was regressed out separately for the left and right hemispheres (Fig. 1b).

Figure 1c and Supplementary Fig. S1 show the mean residual intensity maps. The left-right asymmetry index (AI) was calculated by subtracting the right hemisphere from the left hemisphere. The overall mean map (averaged across layers) showed left-right AI from the anterior to posterior direction (Fig. 1d), indicating that the left hemisphere showed higher cell staining intensity in anterior regions but lower intensity in posterior regions, compared to the right hemisphere. The rightward anchor (most rightward region) was located at the AUD and the leftward anchor was located at the LAN at the network level. The LAN also exhibited strong leftward asymmetry in superficial layers, but became rightward asymmetric from layer III onwards, with a peak in layers IV and VI (Fig. 1e). Regarding the entire cortex, superficial layers showed anterior-posterior asymmetry and deep layers showed inferior-superior asymmetry (Supplementary Fig. S1). To summarize the more left- or right-ward asymmetry along six layers (60 surfaces), we calculated the skewness of AI (Fig. 1f). Skewness overall indicates the difference in intensity as a function of cortical depth. A higher skewness indicates that deep surfaces have a higher intensity relative to superficial surfaces, whereas a lower skewness indicates a less steep difference between upper and lower layers. A high left-right asymmetry of skewness indicates that the intensity distribution is more skewed on the left relatively deeper surfaces. Skewness of left-right asymmetry tells us the distribution of asymmetry across the layers with a high skewness score, indicating that left-right asymmetry shifts on the deeper cortical layers. Skewness differentiation was observed between more leftward asymmetry in somatomotor and more rightward asymmetry in auditory networks.

### Translation to microstructure-sensitive in vivo MRI (Fig. 2)

After establishing asymmetry in cytoarchitecture using a *post-mortem* sample, we aimed to extend the work by using in vivo proxies to capture microstructural differentiation in the cortex. Specifically, we extracted intensity data from HCP T1w/T2w maps (n = 1101) and summarized them using a multimodal parcellation and the CA network atlas. T1w/T2w intensity ranges were homogenized by z-scoring intensity values, vertex-wise, independently for each hemisphere (Fig. 2a).

In this in-vivo sample, we observed a populational asymmetry (using Cohen's d) pattern from anterior to posterior (Fig. 2b). At the network level, the rightward effect anchors were located at Vis2 (Cohen's d = −2.57, $P_{FDR} < 0.001$) and DAN (Cohen's d = −2.19,

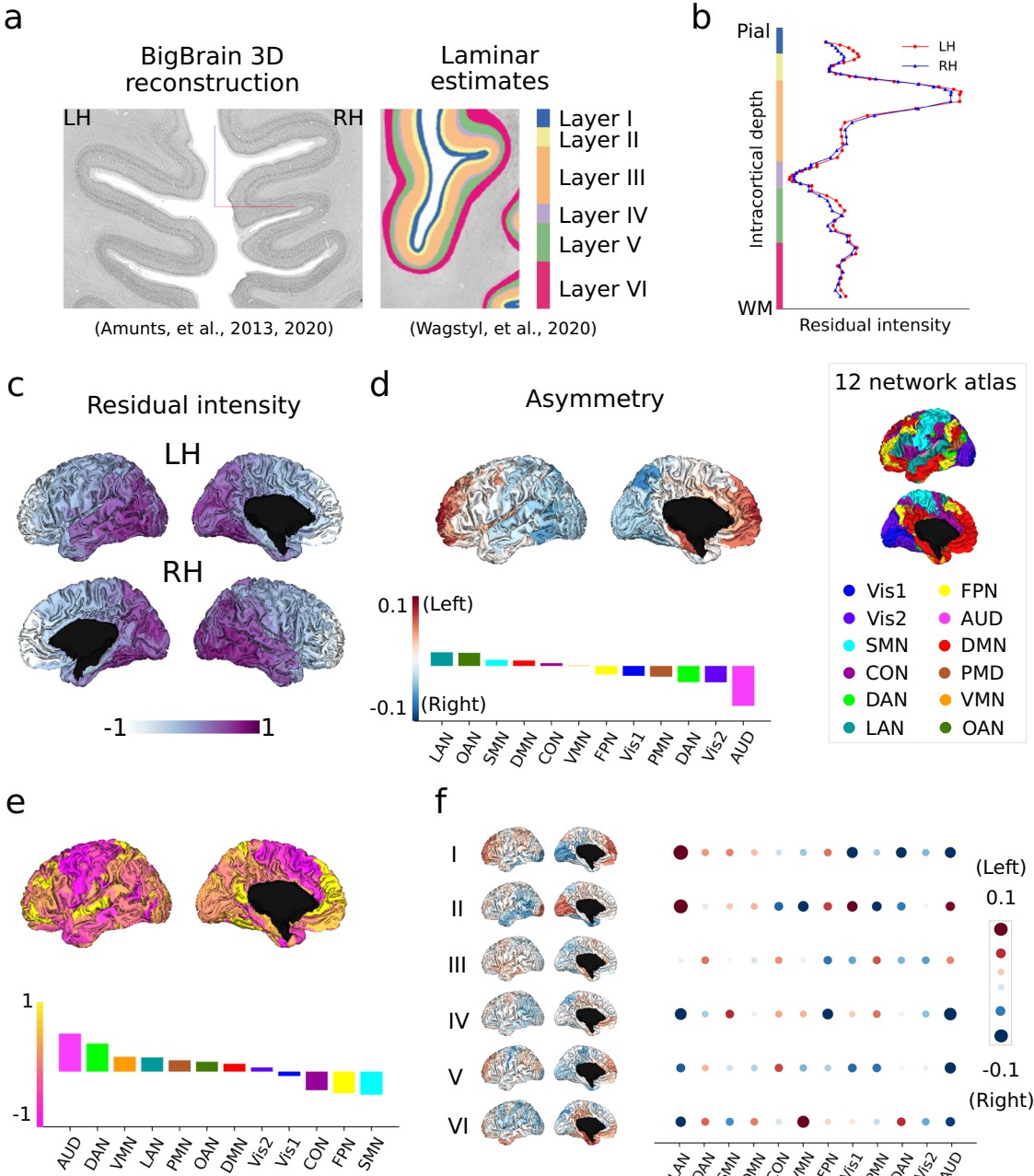

**Fig. 1 | Microstructural asymmetry in BigBrain using cytoarchitecture ($N=1$).**
**a** BigBrain 3D histological reconstruction[15,53] and six-layer estimates[42]. Source: https://bigbrainproject.org/maps-and-models.html. **b** Intracortical staining intensity profiles. Red and blue lines indicate the left and right hemispheres, respectively. **c** Mean intensity maps across 6 layers. **d** Mean asymmetry across layers. Red and blue indicate asymmetry index (AI) left > right and right > left. **e** Layer-wise AI for Bigbrain: six-layer parcel-wise AI brain maps and network-wise heatmap. **f** Skewness map across asymmetry along 60 points of intracortical depth. Atlas-defined networks include primary visual (Vis1), secondary visual (Vis2), somato-motor (SMN), cingulo-opercular (CON), dorsal attention (DAN), language (LAN), frontoparietal (FPN), auditory network (AUD), default mode (DMN), posterior multimodal (PMN), ventral multimodal (VMN), orbito-affective (OAN).

$P_{FDR} < 0.001$); the leftward effect anchors were situated at FPN (Cohen's d = 2.53, $P_{FDR} < 0.001$) and LAN (Cohen's d = 2.13, $P_{FDR} < 0.001$). For networks' effect sizes of the left-right asymmetry see Source Data. Additionally, we observed a similarity in spatial patterns between *post-mortem* cytoarchitectural (BigBrain) and in-vivo micro-structure (HCP) asymmetry ($r = 0.482$, $P_{variogram} = 0.007$, Fig. 2c). Upon further analysis of each layer, we found that in particular layer III exhibited a strong similarity between the two asymmetry maps ($r = 0.513$, $P_{variogram} < 0.001$). Overall, significant correlations were situated in layer I-IV, but not in layer V and VI.

Following this, and building on the twin-pedigree design of the HCP sample, we calculated the heritability ($h^2$) of asymmetry (Fig. 2d).

The three most heritable networks were: Vis2 ($h^2 = 0.51$, SE = 0.05, $P_{FDR} < 0.001$), DAN ($h^2 = 0.43$, SE = 0.05, $P_{FDR} < 0.001$), and FPN ($h^2 = 0.39$, SE = 0.06, $P_{FDR} < 0.001$), see Source Data. The spatial correlation between absolute AI score and heritability maps was Pearson $r = 0.469$ with $P_{variogram} = 0.001$, indicating that regions that are more asymmetric are also more heritable.

Last, to probe potential markers of individual variation, we studied the effects of sex, based on a self-reported question of assigned sex at birth, and age on microstructural asymmetry (Fig. 2e). Both the sex and age $t$-maps revealed an anterior to posterior direction and were correlated with the mean in vivo AI map ($r_{sex} = 0.961$, $P_{variogram} < 0.001$; $r_{age} = 0.690$, $P_{variogram} < 0.001$). These findings

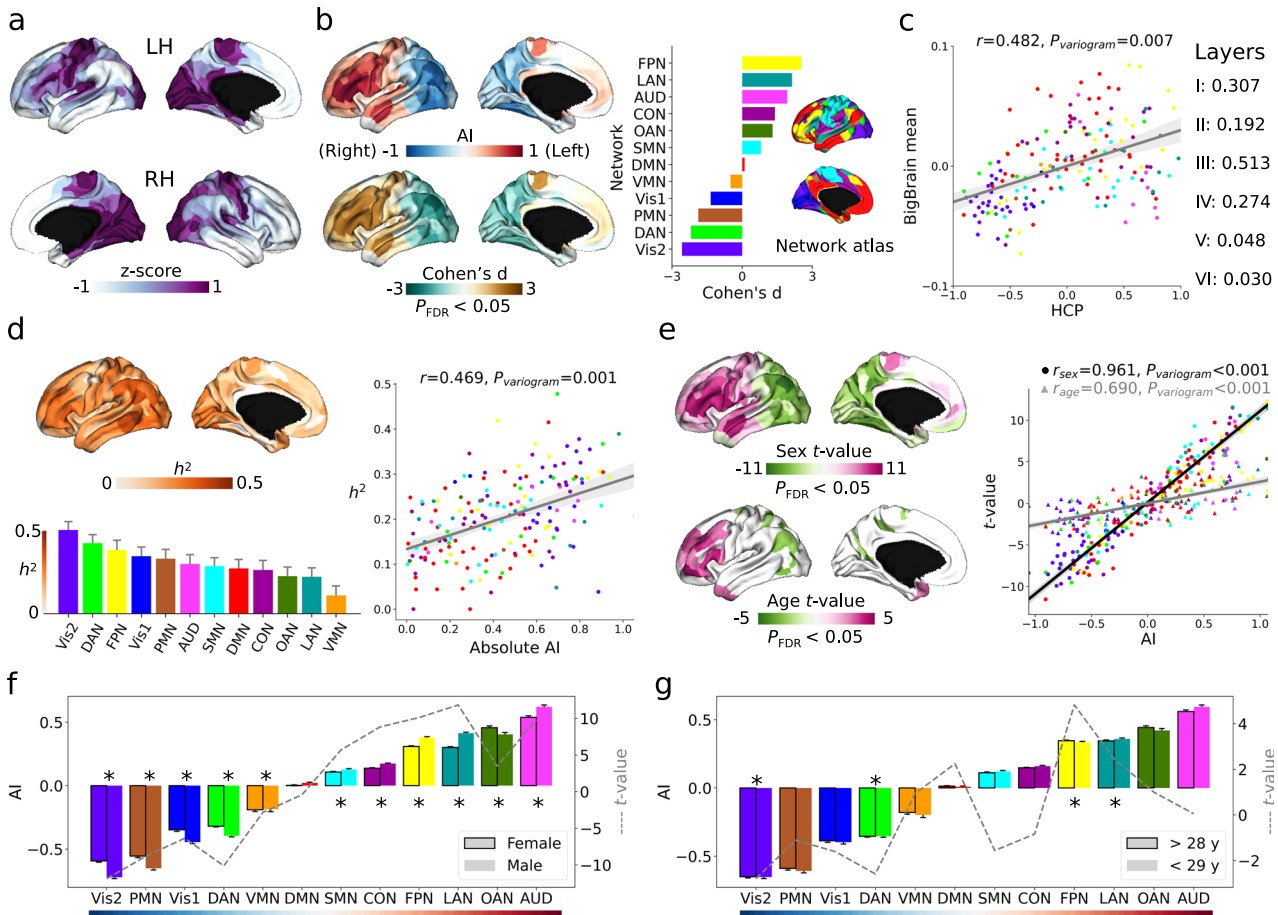

**Fig. 2 | Microstructural asymmetry in Human Connectome Project (HCP) using T1w/T2w images ($N$ = 1101). a** T1w/T2w intensity values for left and right hemispheres (Z-scored separately). Deeper purple indicates higher intensity. **b** The mean asymmetry index (AI) and related Cohen's d maps calculated across subjects. Red/brown and blue/green indicate left- and right-ward asymmetry direction at populational level. AI was also summarized into functional networks with mean and standard error in the barplot. Cohen's d map was thresholded at $P_{FDR} < 0.05$ (two-sided). **c** Spatial correlation between mean HCP AI and BigBrain AI maps. A variogram permutation test was used to account for the spatial autocorrelation. Bold correlation with layer asymmetry indicates significance after variogram permutation at $P < 0.05$ level (two-sided). **d** Heritability map and network barplot (bar is standard error) estimated by individual variation of AI. The heritability map was thresholded at $P_{FDR} < 0.05$ for multiple comparisons correction. Right panel is the spatial correlation between mean AI and heritability maps. **e** T-maps of sex and age effects in the model of AI = 1 + sex + age. Purple red indicates higher leftward asymmetry in females and in older people, respectively. Right panel is the spatial correlation between mean AI and t-value maps. Round and triangle dots represent sex and age. The t-maps were thresholded at $P_{FDR} < 0.05$ for multiple comparisons correction. **f** and **g** plot the detailed sex and age effects in functional networks (AI mean and standard error). Dashed lines indicate t-value for sex and age, respectively. * indicates statistical significance after multiple comparisons ($P_{FDR} < 0.05$). The colors of dots and bars in all plots reflect atlas-defined functional networks including primary visual (Vis1), secondary visual (Vis2), somatomotor (SMN), cingulo-opercular (CON), dorsal attention (DAN), language (LAN), frontoparietal (FPN), auditory network (AUD), default mode (DMN), posterior multimodal (PMN), ventral multimodal (VMN), orbito-affective (OAN).

suggest that microstructural intensity is more asymmetric in males and younger individuals, in this relatively young sample. Details for comparisons in functional networks are shown in Fig. 2f and g. For convenient visualization, we divided age into two groups (i.e., >28 years and <29 years) but t-values were reported based on continuous age. There were 11 out of 12 networks statistically significant after FDR correction for sex comparisons, excluding only DMN ($t = −0.422$, $P_{FDR} = 0.673$). There were 4 out of 12 networks statistically significant for age comparisons, including Vis2 ($t = −2.797$, $P_{FDR} = 0.031$), DAN ($t = −2.589$, $P_{FDR} = 0.039$), FPN ($t = 4.826$, $P_{FDR} < 0.001$), and LAN ($t = 2.456$, $P_{FDR} = 0.042$).

**Microstructural asymmetry is linked with asymmetry in intrinsic function (Fig. 3)**

After establishing microstructural asymmetry in both *post-mortem* and in vivo markers, we investigated its functional association. To achieve this, we utilized resting state functional connectivity (FC) in the same sample (i.e., HCP), as in previous research[6].

To investigate the relationship between microstructure and function asymmetry at the group level, we divided the mean microstructural asymmetry map into 10 bins (see Fig. 3a-i). We then calculated the average functional connectivity (FC) within each bin (see Fig. 3a-ii) and determined the functional connectivity asymmetry by subtracting the right hemisphere (RH) from the left hemisphere (LH) and dividing by the sum of RH and LH. Our analysis focused on intra-hemispheric (i.e., LH_LH and RH_RH) connectivity. Functional connectivity was observed to be stronger between regions that exhibited similar patterns of asymmetry, compared to those with varying degrees of asymmetry (Fig. 3a-iii). This relationship was quantitatively assessed by calculating region-wise microstructure-function correlation between T1w/T2w AI map and FC AI profile (Fig. 3b-i). We found coupling was strongest in central and superior temporal areas and weakest in prefrontal and parietal areas, both at the group level and individual level ($r = 0.664$, $P_{variogram} < 0.001$). As demonstrated in the map of mean and standard deviation, this coupling exhibited significant individual variability, particularly in areas of overall strong coupling (Fig. 3b-ii).

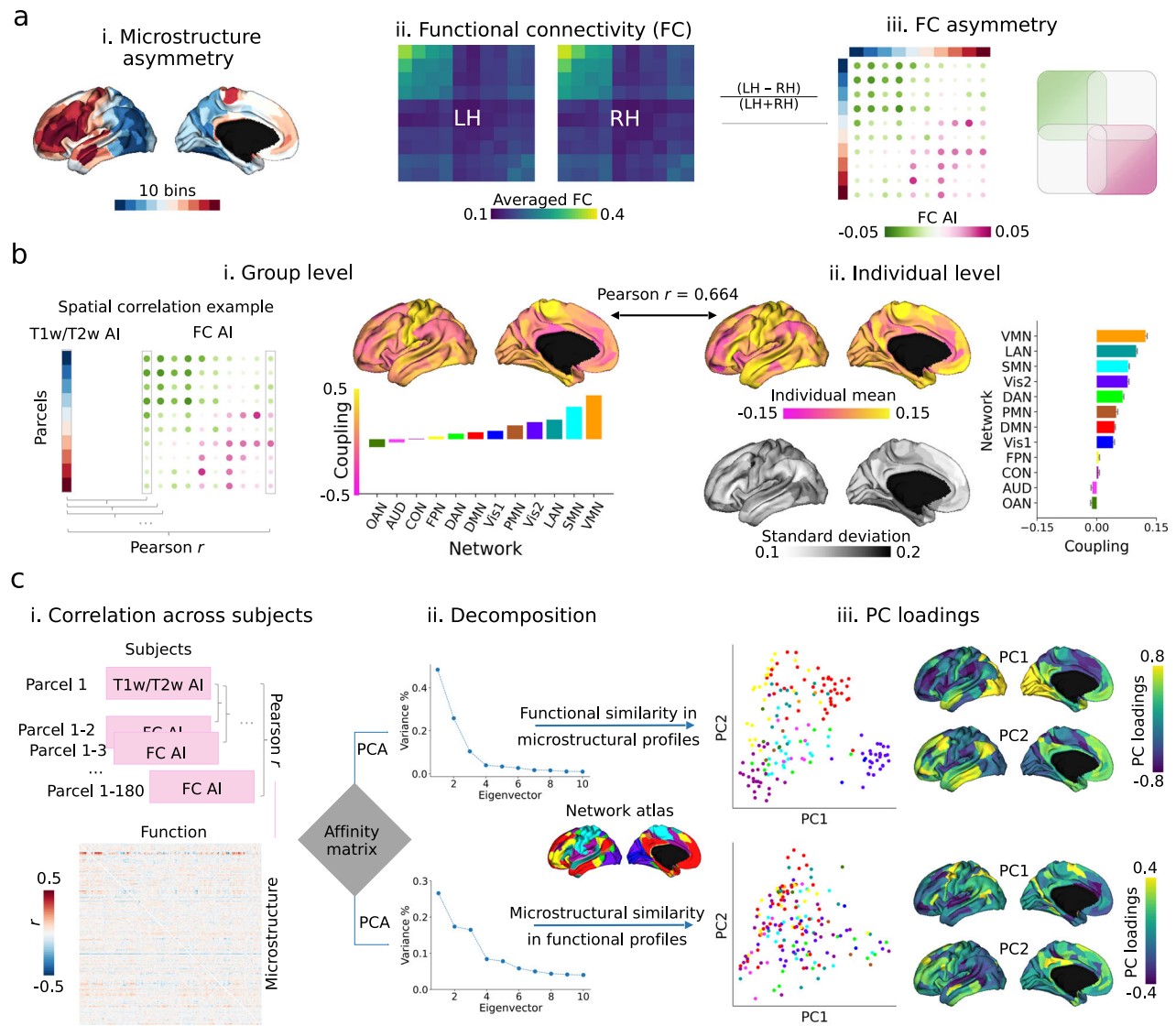

**Fig. 3 | Microstructure-function relationship in asymmetry using HCP T1w/T2w and resting state functional images (N = 1004). a–i** 10 bins (18 parcels per bin) categorized from Fig. 2b mean AI map (T1w/T2w). **a-ii.** Group-level resting state functional connectivity (FC) matrix averaged by bins. **a-iii.** FC asymmetry calculated by (LH - RH)/(LH + RH) sorted by bins. Purple-red and green indicate left- and right-ward asymmetry. Scatters are colored by functional networks. **b** Region-wise microstructure-function coupling was calculated by Pearson correlation coefficient between 180 parcels of T1w/T2w and FC AI per column. Left panel shows coupling between mean maps at the group level (i) and the right panel shows mean and standard deviation of coupling (ii). **c** Individual covariation between microstructure and function. Matrix in

(i) represents the Pearson r between parcel T1w/T2w AI and FC AI across subjects. Then, the parcel-wise affinity matrix was computed and principal component analysis (PCA) was employed to decompose the matrix to detect the inter-region similarity axes (ii). Upper and lower panels are microstructural and functional decomposition (iii). The first two eigenvectors and eigenvalues (PC loadings) are plotted with similar colors in 'viridis' indicating similar profiles between regions. Atlas-defined networks include primary visual (Vis1), secondary visual (Vis2), somatomotor (SMN), cingulo-opercular (CON), dorsal attention (DAN), language (LAN), frontoparietal (FPN), audi-tory network (AUD), default mode (DMN), posterior multimodal (PMN), ventral multimodal (VMN), orbito-affective (OAN).

Finally, to study how asymmetry in microstructure-function coupling varied between regions, we calculated the individual co-variation between microstructural and FC asymmetry per parcel across subjects (Fig. 3c-i). Then we extracted the top 10% of the co-variation to calculate the affinity matrix using a normalized angle. Finally, using principal component analysis (PCA), we decomposed the affinity matrix by rows and columns. Row PCs summarized microstructural similarity in functional profiles and column PCs summarized functional similarity in microstructure (Fig. 3c-ii). For the microstructural PCs, the first two components accounted for 26.6% and 17.4% of the total variance, respectively. PC1 displayed a dissimilarity axis from the dorso-lateral prefrontal to the precentral gyrus, while PC2 displayed a dissimilarity axis from the temporoparietal junction to the lateral

prefrontal areas. Regarding the functional PCs, the first two components accounted for 48.4% and 25.8% of the total variance, respectively. PC1 differentiated prefrontal from visual regions, and PC2 differentiated sensory from association regions (Fig. 3c-iii).

## Microstructural asymmetry relates to individual variability in language skills and mental health (Fig. 4)

Our last aim was to investigate the behavioral relevance of micro-structural asymmetry. Language scores were obtained through read-ing and picture vocabulary tests. Mental health scores were assessed using the Adult Self-Report and DSM-Oriented Scale, which included depression, anxiety, somatic, avoidant, ADHD, and antisocial pro-blems. Therefore, two variables for language, six variables for mental

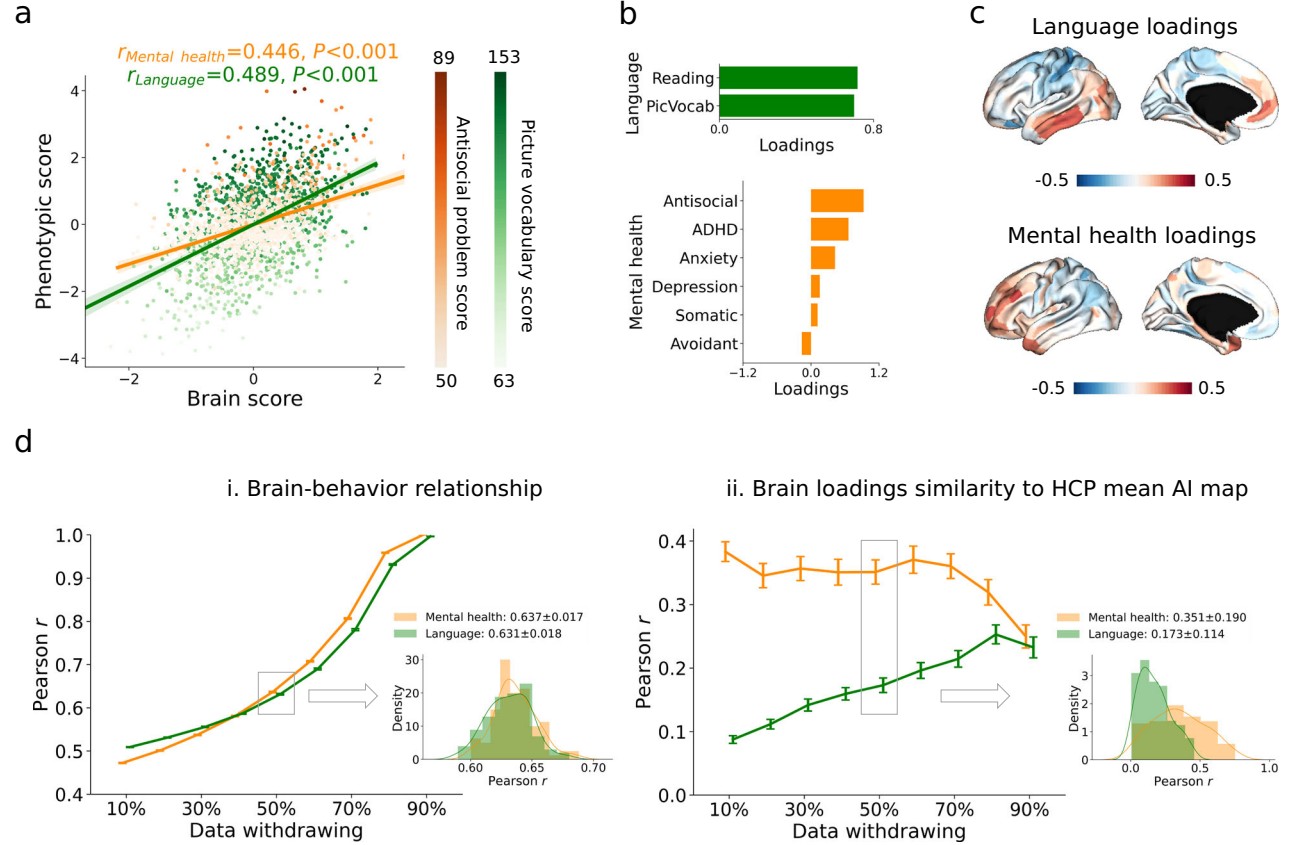

**Fig. 4 | Canonical correlation analysis (CCA) between microstructural asymmetry features and language/mental health in HCP. a** Correlation between latent dimensions of the brain and phenotype. Orange and green indicate mental health and language, where the latent dimensions explain antisocial behavior and picture vocabulary scores most. **b** Phenotypic loadings of the first latent dimension for language and mental health. **c** Brain loadings of the first latent dimension for language and mental health. **d** Resampling data to test the performance of CCA. We withdrew data from 10% to 90% by pseudo-randomization using twin classes and resampled 100 times. Mean and standard error bars were shown in the charts. Data withdrawal of 50% was selected to show the distribution across the 100 samples and other percentages see Supplementary Fig. S4. PicVocab: picture vocabulary; ADHD: attention deficit/hyperactivity disorder.

health, and 180 brain variables (regional microstructural asymmetry) were included. To achieve this, we conducted a canonical correlation analysis (CCA), which is a multivariate approach to estimate latent dimensions for multiple independent and dependent variables via correlation.

We calculated the first latent dimension for language and mental health separately. Statistically significant correlations were found between the first latent dimensions of microstructural asymmetry and behavioral markers ($r_{language} = 0.489$, $P < 0.001$; $r_{mental\ health} = 0.446$, $P < 0.001$, Fig. 4a). Picture vocabulary and antisocial problem scores had the strongest loadings on the respective brain-behavior latent components (Fig. 4b). Moreover, we observed a behavioral marker divergence of spatial patterns in brain loadings between language and mental health. Whereas the former showed a differentiation between superior and inferior areas, mental health was linked to differentiation between anterior and posterior portions in asymmetry, with stronger leftward asymmetry in frontal regions associated with reduced mental health (higher scores) (Fig. 4c). The anterior-posterior layer-AI maps were similar to mental health brain loadings ($r_{HCP} = 0.415$, $P_{variogram} = 0.024$; $r_{layer\ III} = 0.315$, $P_{variogram} = 0.020$, Supplementary Fig. S2) but the more inferior-superior layer-AI map was similar to language brain loadings ($r_{layer\ V} = 0.210$, $P_{variogram} = 0.061$, Supplementary Fig. S3). This may suggest a reduced leftward asymmetry in the frontal lobe and an increased rightward asymmetry in the occipital lobe for individuals with higher mental health scores.

To test the robustness of the findings, we conducted the pseudo-randomized resampling, 100 times, withdrawing 10%-90% of the data

based on the twin label (Fig. 4d). We show the average result (50% of data withdrawn) in the main figure and the remaining in Supplementary Fig. S4a and b. We found that the first latent brain and behavioral dimensions remained robust ($r_{language} = 0.631 \pm 0.018$, $r_{mental\ health} = 0.637 \pm 0.017$). The correlation between mean AI map and brain loadings were $r_{language} = 0.173 \pm 0.114$ and $r_{mental\ health} = 0.351 \pm 0.190$.

## Concordant validation

To replicate our findings using independent samples, we conducted in vivo analyses on two external datasets with varying imaging measurements for cortical microstructure. These datasets include MICs in young adults (qT1 relaxometry, $N = 50$) and the NSPN in adolescents and young adults (MT, baseline $N = 286$).

Regarding MICs qT1 relaxometry data, the sample demographics resembled HCP data (female: 46%, age centered on 25-35 years old). An anterior-posterior asymmetry pattern was again observed (Supplementary Fig. S5a-e) with a significant correlation between MICs and the HCP AI map ($r = 0.548$, $P_{variogram} = 0.004$). The spatial pattern of the sex effect was also replicated ($r_{sex} = 0.731$, $P_{variogram} < 0.001$) whereas the age effect was not ($r_{age} = 0.224$, $P_{variogram} = 0.062$). We used MT images using multi-parametric mapping[56] from NSPN to address potential transmit field issues associated with T1w/T2w. The sample consisted of individuals aged 14 to 25 years (mean ± SD: 19.1 ± 2.9) with a balanced sex ratio (female: 51%). Again we observed the anterior-posterior asymmetry pattern and found a significant correlation ($r = 0.369$, $P_{variogram} = 0.047$) between HCP and NSPN (Supplementary Fig. S6a-e). In addition, the spatial pattern of age and sex effects could also be

replicated in this younger sample ($r_{sex} = 0.358$, $P_{variogram} < 0.001$; $r_{age} = 0.323$, $P_{variogram} < 0.001$). Together, these analyses suggest that microstructural asymmetry is consistent across different microstructural measures and histology.

To further test the robustness of our findings, we also used a raw intensity score to calculate the asymmetry index by (LH - RH)/(LH + RH). Overall findings were consistent but showed a subtle difference (all spatial Pearson $r > 0.9$) in the medial frontal cortex for asymmetry, sex and age $t$-maps (Supplementary Fig. S7). In addition, we tested for potential association between handedness and microstructural asymmetry. We found no parcels that survived statistical thresholds.

## Discussion

Asymmetry in structural and functional brain organization is implicated in key human cognitive functions, including language, and is associated with neuropsychiatric conditions[1–7]. In this study, we used a multiscale approach to investigate microstructural asymmetry at ultra-high resolution and applied our model to in vivo data to examine individual variability and functional relevance. A consistent pattern of left-right asymmetry of microstructure, along the anterior to posterior regions, was observed in an ultra-high-resolution *post-mortem* human brain and three in vivo samples. Using an ultra-high resolution *post-mortem* model, we found that the asymmetry pattern differed across layers, with superficial layers showing an anterior-posterior pattern and deep layers showing an inferior-superior pattern. Furthermore, utilizing an in vivo model of microstructure, we demonstrated that microstructural asymmetry varies with age and self-reported sex and is heritable, using a twin model. Finally, we established the functional relevance of these findings by linking microstructural asymmetry to asymmetry in intrinsic functional connectivity profiles as well as behavioral markers that detail individual variations in language skills and mental health traits. This can be viewed as a proxy of neuropsychiatric risk within the healthy population. Language skills vary along an inferior-superior axis, while mental health traits vary along an anterior-posterior axis, suggesting that mental health may be associated with superficial laminar functions but language may coordinate with deep laminar functions, a divergence possibly anchored in neurodevelopmental trajectories of both patterns. Together, our findings illustrate consistent cortical brain asymmetry in cytoarchitecture and microstructure and its functional correlates.

In the current work we estimated asymmetry of human cortical cytoarchitecture and uncovered an overall pattern of left- to rightward asymmetry in an anterior-to-posterior axis. Previous studies on regional cytoarchitecture have indicated that the left hemisphere has a higher neuronal density in inferior frontal cortex, i.e., Brodmann area (BA) 45[7] and dorsolateral prefrontal cortex, i.e., BA 9. These have been attributed to pyramidal neurons in layer III[57]. Leftward asymmetry was also reported in anterior regional torque for Broca's area and rightward asymmetry was reported in posterior regional torque for occipital visuospatial area[4]. The anterior-posterior differentiation in asymmetry may be related to neurodevelopmental patterning and cortical maturation. Indeed, whereas posterior regions show early postnatal development, anterior regions mature in adolescence, illustrated by variations of cyto- and myelo-architecture and connectivity[58]. Though the current study did not evaluate developmental patterning over time, and the subject (the BigBrain data) was also older than 60 years, it is possible that the asymmetry observed is still a consequence of maturational timing in combination with experience-dependent plasticity due to the differential functional role of the left and right hemisphere.

Moreover, we observed notable depth-wise variation with the anterior-posterior asymmetry in upper and inferior-superior asymmetry in deep layers. Overall, maturation patterns of microstructure during childhood have been shown to follow a posterior-anterior pattern[59]. Previous work has reported divergent maturational profiles

in intra-cortical microstructure between sensory and paralimbic areas in adolescence[60]. In particular, mid-to-deep layers seem to have a preferential development in adolescents, specifically in uni- and hetero-modal areas spatially corresponding to attention and language regions[60]. In our work, the language network shifted from leftward (superficial layers) to rightward asymmetry (deep layers), which may be related to the development and maturation of cortico-cortical connections[61–63]. The laminar architecture of the cortex has an important role in coordinating functional processes and connectivity between cortical regions and subcortex and cortex[64]. For cortico-cortical or cortico-subcortical pathways, cellular and synaptic architectures differ across layers such that they result in distinct computations at the target projection neurons[65]. Though observing asymmetry across layers results in novel hypotheses and perspectives on the neuroanatomical origin of functional specializations in the cortex, indicating asymmetry not only varies spatially along the cortical mantle but also as a function of its depth.

Microstructural asymmetry in the anterior-posterior direction could also be identified using in vivo imaging, using T1w/T2w maps, qT1 relaxometry, and MT. Previous work has suggested that the anterior-posterior asymmetry pattern in T1w/T2w is partly generated by the transmit field rather than the microstructure itself[8]. However, T1w/T2w images have been corrected for some of the B1+ bias (see Methods), and using flip angles map to further correct the image might reduce the real signals[8]. In addition, we used qT1 and MT to validate our in vivo results and observed concordance across all metrics. Yet, qT1 and MT contrast used in the current study are likely less affected by the transmit field, suggesting that at least a part of the effects may go above and beyond signal noise. In addition, although all MRI measurements are sensitive to myelination, they still have differences[66]. Theoretically, MT and qT1 detect exchange and cross-relaxation between lipids and water in tissue, and T1w/T2w means to correlate based on neurobiological principles by contrast of fitting the decay curve using least-square methods in non-myelin water pool[67]. Compared to MT and qT1, T1w/T2w includes more non-myelin signals. Taking advantage of the inter-individual model of in vivo data, we probed its relevance to age and sex in the samples tested. We found that more asymmetric regions overall show a stronger age effect. Increased cortical symmetry with age has been previously reported in the mid- and old-aged sample of the UK Biobank[68] and young adults[69]. Looking ahead, the implementation of a normative model could elucidate the developmental trajectory of microstructural asymmetry across the lifespan, facilitating the identification of individual trajectories[70]. Previous work has also reported marked sex differences in various indices of cortical structure[71–74], possibly related to differential sex hormonal expression and physiological markers[75]. Indeed, in the current work we observed an overall stronger asymmetry in males relative to females. Related work reported overall higher mean microstructure and lower skewness in males relative to females, a difference that varied as a function of (self-reported) hormonal status in females[74]. However, in the current work we could only touch upon these neuroendocrine, physiological, and age-related factors shaping microstructural brain asymmetry. Further work, including hormonal data and broader age ranges may further reveal potential causes and consequences of the observed associations between age and self-reported sex and brain asymmetry. Of note, in the current work, we found no association between handedness and microstructural asymmetry. Other work either reports little association between handedness and cortical thickness and surface area in the multi-center ENIGMA data[10]. However, handedness and polygenic risk scores of handedness are associated with macrostructural asymmetry in a few regions in the UK Biobank[76]. Future studies may focus on meta-analysis to identify whether handedness is associated with microstructural asymmetry with more papers published or use more fine-grained investigations of dexterity and brain anatomy.

We found that microstructure and intrinsic function have similar lateralized directions within hemispheres. In the cerebral cortex, regions with similar cytoarchitecture tend to have similar functional connectivity profiles[16,34]. We showed that this principle may also hold for asymmetry of microstructure and functional connections. In particular, somatomotor and language-related areas show an increased convergence between functional and microstructural asymmetry profiles. These regions have highly specialized functions, possibly suggesting that functional specialization may be accounted for by cortical asymmetry[14,77,78]. Relatedly, via inter-individual co-variation, we observed that regions within functional networks share more similarity in microstructural asymmetry profiles, again underscoring the link between microstructural asymmetry and intrinsic function. This pattern could be interpreted as suggesting that activity-dependent plasticity in part shapes the microstructural asymmetry observed, ultimately supporting similar microstructural profile asymmetry in functionally connected regions. At the same time, the inverse of the model, i.e. microstructure-guided function, revealed clearer organizational patterns.

Probing functional relevance of microstructural asymmetry in terms of behavioral outcomes, we observed a pattern of inferior-superior differentiation for language. This was mainly linked to a differentiation of temporal lobe and sensorimotor regions, which spatially mirrored the asymmetry pattern in layer V of the BigBrain. Layer V is the main output layer of the cortex and largely relays signals to subcortical structures[64], yet also shows marked divergence of connection profiles as a function of neuron type[79]. On top of this, we found that cell density is lower in the right relative to the left hemisphere for the language network in the BigBrain. Thus, the observed patterns possibly relate to a differentiation of cells in deeper cortical layers (temporal and sensorimotor) linked to differential output profiles ultimately leading to behavioral differences in language. Second, we found a behavioral marker of mental health to be associated with anterior-posterior differentiation in asymmetry, a pattern present in superficial layers of the BigBrain, and in overall maturational patterns of microstructure in the cortex[59]. Various studies have reported associations between microstructure and neuropsychiatric conditions, including depression[80–82], compulsivity and impulsivity[83], and schizophrenia[84–86]. Through large sample size investigation, it would be possible to study the inter-relationships among asymmetries of cortical brain maturation and neurodevelopmental conditions, extending current work on maturational differentiation of symmetric microstructural patterning and links to disease progression[84,87].

While our research has yielded significant insights into cortical microstructural asymmetry, it is important to address several limitations that warrant clarification. Firstly, though the BigBrain (N = 1) offers unique insights into cortical microstructure at ultra high resolution, and links to our in vivo model, the results are limited to one subject. Further work on ultra-high resolution neuroimaging (e.g., 7 T or 9.4 T MRI), sensitive to laminar changes, will aid in also understanding layer-level markers of individual variation. Although T1w/T2w images reveal a strong anterior-posterior asymmetry pattern and concordant validation from other microstructural measures have been found, it still requires more B1+ bias correction to reduce the inhomogeneities in HCP T1w/T2w. Furthermore, while the observed map of age effects in our young adult sample correlated with the mean asymmetry map, no parcels exhibited significant age effects in the MICs and NSPN datasets. The modest changes observed during adolescence may stem from either small effect sizes or inadequate sample sizes to detect statistical significance. Additionally, it is crucial to acknowledge that the mental health data utilized in this study pertains solely to healthy individuals, and caution should be exercised in extrapolating these findings to clinical samples, where extreme conditions may prevail that were not accounted for in our study.

In conclusion, our investigation employed a multiscale approach to study microstructural asymmetry in the human cortex. We delineated laminar-specific asymmetry in a *post-mortem* sample and individual asymmetry in vivo. Our study contributes to advancing our understanding of cortical asymmetry at the microscale, encompassing depth-wise and inter-regional spatial differentiation, age and sex disparities, behavioral genetics based on twin-modeling, integration with functional connectomics, and associations with behavioral markers of language and mental health. These findings hold implications for elucidating the biological mechanisms underlying cortical asymmetry and its functional relevance in health and disease.

## Methods

Datasets we used in present study are open sources and have been approved by their local research ethics committees. The current research complies with all relevant ethical regulations as set by The Independent Research Ethics Committee at the Medical Faculty of the Heinrich-Heine-University of Duesseldorf (study number 2018–317). The specifics on the MRI data and methods applied are the similar to related works in the same samples. They are provided again here for completeness.

### Datasets & image acquisition and preprocessing

**BigBrain.** BigBrain is a 20 μm³ ultra-high-resolution atlas of a *post-mortem* human brain from a 65-year-old male created by digital volumetric reconstruction of Merker-stained sections (https://ftp.bigbrainproject.org/)[15]. The six layers of BigBrain's cerebral cortex were previously segmented using a convolutional neural network and the surface reconstruction of the layer boundaries were available[42]. We extracted layer-wise cortical profiles of the BigBrain cerebral cortex by sampling the staining intensity of 100 μm resolution BigBrain images at 10 equivolumetric surfaces along the depth of each layer (60 surfaces in total). The resulting layer-wise cortical profiles reflect the variation of neuronal size and density along the depth of the six cortical layers at each location.

To reduce the computational demands, we downsampled the images from BigBrain native surface space to Glasser multimodal parcellation[54], a homologous atlas with 180 parcels per hemisphere via BigBrainWarp[88]. To enhance the functional annotation we employed the cortical functional network atlas[55], which includes 12 networks: primary visual (Vis1), secondary visual (Vis2), somatomotor (SMN), cingulo-opercular (CON), dorsal attention (DAN), language (LAN), frontoparietal (FPN), auditory network (AUD), default mode (DMN), posterior multimodal (PMN), ventral multimodal (VMN), and orbito-affective (OAN).

**HCP.** We used T1w/T2w images from the Human Connectome Project (HCP) S1200 release, which can be downloaded from HCP DB (http://www.humanconnectome.org/). HCP S1200 includes 1206 individuals (656 females) that are made up by genetic-identified and reported 334 MZ twins, 152 DZ twins, and 720 singletons. We included individuals for whom the scans and data had been released after passing the HCP quality control and assurance standards[89,90]. Finally, for genetic analyses we included 1101 healthy subjects with a good quality T1w/T2w image (age: 28.8 ± 3.7 years), of which 54.4% were females and 332 were MZ twins.

MRI data were acquired on the HCP's custom 3 T Siemens Skyra equipped with a 32-channel head coil. Two T1w images with identical parameters were acquired using a 3D-MP-RAGE sequence (0.7 mm isovoxels, matrix = 320 × 320, 256 sagittal slices; TR = 2400 ms, TE = 2.14 ms, TI = 1000 ms, flip angle = 8; iPAT = 2). Two T2w images were acquired using a 3D T2-SPACE sequence with identical geometry (TR = 3200 ms, TE = 565 ms, variable flip angle; iPAT = 2). T1w and T2w scans were acquired on the same day. The pipeline used to obtain the Freesurfer-segmentation is described in detail in a previous article[89]. The preprocessing steps included co-registration of T1- and T2-weighted scans, then correcting the T1w and T2w images for B1- bias

and some B1+ bias[26,89]. Preprocessed images were nonlinearly registered to MNI152 space, and segmentation and surface reconstruction performed using FreeSurfer 5.3. T1w images were divided by aligned T2w images to produce a single volumetric T1w/T2w image per subject[26]. Cortical surfaces were aligned using MSMAll[91,92] to the hemisphere-matched conte69 template[43]. Notably, this contrast nullifies inhomogeneities related to receiver coils and increases sensitivity to intracortical myelin.

The intensity values were estimated between pial and white matter surfaces. Previous papers have used this data to generate the equivolumetric profile intensity[34,93]. We downsampled the images from conte69 space to the multimodal atlas. In this study, we averaged the intensity values across the equivolumetric surfaces and z-scored the values for left and right hemispheres separately for each subject.

**MICs.** For replication analysis we used the quantitative T1 images of the openly available MRI dataset for Microstructure-Informed Connectomics (MICs)[28], which can be downloaded from the Canadian Open Neuroscience Platform's data portal (https://portal.conp.ca). The dataset comprises multimodal data of 50 healthy young adults (23 women; $29.54 \pm 5.62$ years; 47 right-handed) and was collected at the Brain Imaging Centre of the Montreal Neurological Institute and Hospital using a 3 T Siemens Magnetom Prisma-Fit and a 64-channel head coil. For the acquisition of the qT1 relaxometry data a 3D magnetization prepared 2 rapid acquisition gradient echoes sequence was used (3D-MP2RAGE; 0.8 mm isotropic voxels, 240 sagittal slices, TR = 5000 ms, TE = 2.9 ms, TI_1 = 940 ms, T1_2 = 2830 ms, flip angle 1 = 4°, flip angle 2 = 5°, iPAT = 3, bandwidth = 270 Hz/px, echo spacing = 7.2 ms, partial Fourier = 6/8). Two inversion images were combined for qT1 mapping. Based on the varying T1 relaxation time in fatty tissue compared to aqueous tissue[94], we here use qT1 as an index for gray matter myelin and hence as a proxy for microstructure. The MRI processing tool *micapipe*[95] was used for data preprocessing and intensity extraction. In short, preprocessing included the background denoising of MP2RAGE, reorientation of the T1W and MP2RAGE, N4 bias correction, and intensity rescaling of the T1W images and the nonlinear registration to MNI152 space. Further, the cortical surface reconstruction from native T1w acquisitions was carried out using Freesurfer 7.0. The detailed acquisition protocol and preprocessing are described in their prior data publication[28].

**NSPN.** The Neuroscience in Psychiatry (NSPN) cohort generally comprises 2245 adolescents aged 14 to 26 years (mean ± SD age: $19.1 \pm 3.0$ years, female: 54%). Participants were recruited in Cambridgeshire and north London according to a sampling design that balanced sex, ethnicity, and participant numbers in five age strata (14-15, 16-17, 18-19, 20-21, 22-25). Here, we included 286 individuals (mean ± SD age: $19.1 \pm 2.9$ years, female: 51%) for whom microstructural neuroimaging data were available.

Magnetization Transfer (MT) data were acquired to approximate myelin content using a multi-parametric mapping (MPM) sequence[56] on three identical 3 T Siemens MRI Scanners (Magnetom TIM Trio) in Cambridge (2 sites) and London (1 site). A standard 32-channel radio-frequency (RF) receive head coil and RF body coil for transmission were used. MPM included three multi-echo 3D FLASH scans: predominant T1-weighting (repetition time (TR) = 18.7 ms, flip angle = 20°), and predominant proton density (PD) and MT-weighting (TR = 23.7 ms; flip angle = 6°). To achieve MT-weighting, an off-resonance Gaussian-shaped RF pulse (4 ms duration, frequency offset from water resonance = 2 kHz; nominal flip angle = 220°) was applied prior to the excitation. Several gradient echoes were recorded with alternate readout polarity at six equidistant echo durations (TE) between 2.2 and 14.7 ms for MT-weighted acquisition. The longitudinal relaxation rate and MT signal are separated by the MT saturation parameter, creating a semi-quantitative measurement that is resistant to relaxation times

and field inhomogeneities[56,96]. Other acquisition parameters include 1 mm isotropic resolution, 176 sagittal partitions, field of view (FOV) = 256×240 mm, matrix = 256×240×176, non-selective RF excitation, RF spoiling phase increment = 50°, parallel imaging using GRAPPA factor two in phase-encoding (PE) direction (AP), readout bandwidth = 425 Hz/pixel, 6/8 partial Fourier in partition direction. The acquisition time was approximately 25 min. Participants wore ear protection and were instructed to lie still.

Surface reconstruction was carried out based on T1-weighted (T1w) images using Freesurfer 5.3.0. The resulting reconstructions underwent visual inspection. Control points were added to improve segmentations, but scans were excluded in cases of persistently poor quality. The detailed description about this dataset and preprocessing are shown in their previous work[56,60,97,98].

**Asymmetry Index.** We calculated the asymmetry index (AI) by subtracting right from left hemispheric values in the homologous regions. As noted, we preprocessed the left and right hemispheres separately by regressing out mean surface intensity for the BigBrain data, then standardized the residual intensity. For HCP, MICs, and NSPN, we obtained the mean cortical intensity map, then z-scored the map for left and right hemispheres separately. For functional connectivity asymmetry, we calculated AI by (LH - RH)/(LH + RH). Regarding the asymmetry of skewness in BigBrain, the skewness formula was used: skewness = sum((intensity $_{surface}$ − mean)$^3$)/SD$^3$, where mean and SD are calculated across the sixty surfaces.

**Effects of sex and age.** We first used fixed effects estimates for the model: AI = 1 + sex + age + sex*age. We found no significant interaction between age and sex. We then used the non-interaction model: AI = 1 + sex + age to obtain the $t$ and $P$ values of sex and age. False discovery rate (FDR) was then applied for multiple comparison correction for the sex and age $t$ value maps. All the steps were performed in Python with the package BrainStat[99]. Regions colored in the HCP figure survived from FDR correction ($q < 0.05$). We didn't perform FDR correction for NSPN and MICs datasets because too few parcels survived. In comparison, for the $t$ and asymmetry maps between the different datasets we used non-thresholded maps.

We performed the correlations between brain maps using variogram permutations to test the spatial autocorrelation. The variogram quantifies, as a function of distance $d$, the variance between all pairs of points spatially separated by $d$. Pure white noise, for example, which has equal variation across all spatial scales, has a flat variogram (i.e., no distance dependence). Brain maps with very little spatial autocorrelation will therefore have a variogram that is nearly flat. Strongly autocorrelated brain maps exhibit less variation among spatially proximal regions (at small $d$) than among widely separated regions, and are therefore characterized by positive slopes in their variograms[100,101]. We obtained the geodesic distance matrix of the left hemisphere from multimodal parcellation (i.e., 180*180) and produced 1000 permuted spatial autocorrelation-preserving surrogate brain maps whose variograms were approximately matched to a target brain map's variogram.

**Heritability analysis.** Referring to our previous work[6,93,102,103], we analyzed heritability based on the twin design of HCP. Briefly, we calculated the heritability estimates with standard errors via the Sequential Oligogenic Linkage Analysis Routines (SOLAR, version 9.0.0). SOLAR uses maximum likelihood variance decomposition methods to determine the relative importance of familial and environmental influences on a phenotype by modeling the covariance among family members as a function of genetic proximity[104]. Heritability, i.e., narrow-sense heritability $h^2$, represents the proportion of the phenotypic variance ($\sigma^2_p$) accounted for by the total additive genetic variance ($\sigma^2_g$), that is $h^2 = \sigma^2_g/\sigma^2_p$. Phenotypes exhibiting stronger covariances between genetically more similar individuals, than between genetically less

similar individuals, have higher heritability. In this study, we quantified the heritability of asymmetry of functional gradients using the A + E model as suggested by prior study, as the A + E model has a higher accuracy of estimating heritability than the A + E + C (common environment) model in HCP[105]. We also included the A + C + E model in Supplementary Fig. S11. We added age, sex, age$^2$, and age*sex as the covariates to our models.

**Microstructure-function asymmetric coupling.** We conducted three approaches to understand the relationship between microstructural and functional asymmetry. We first split the mean microstructural asymmetry map into 10 bins and plotted the FC asymmetry along the 10 bins to test whether group level FC asymmetry was stronger or weaker in asymmetric or non-asymmetric bins. Second, we correlated the microstructural asymmetry with the FC asymmetry spatially at the group and individual levels. In particular, the correlation coefficient was computed between microstructural asymmetry (in 180 regions) and the (180*180) seed-based functional asymmetry map resulting in a regional coupling score. Following, a map of regional coupling between microstructure and function (180 $r$ values) was obtained for all 180 functional seeds. Regarding the covariation across subjects, for a given region, we did the following: (region x, one microstructural asymmetry marker) and (region x, 180 functional asymmetry markers of its connectivity), were correlated along the "region x" axis to obtain 180 r values for this region, indicating how this region's microstructural asymmetry supports asymmetry of functional connection to this region across subjects. This procedure was repeated for all 180 parcels. The 180*180 covariation matrix can be obtained with the columns as microstructural profiles and rows as functional profiles. Then, principal component analysis (PCA) was used to decompose the affinity of the covariance matrix with the sparsity of top 10% scores. These principal components reflect the organization features of asymmetric microstructure-function coupling, e.g., two regions having similar asymmetric coupling profiles get close loadings along the PCs. These steps were done in Python with the package BrainSpace[101].

**Brain-behavioral association.** We used canonical correlation analysis to address the multivariate association between microstructural asymmetry and behavioral scores. CCA is a statistical method that finds linear combinations of two random variables so that the correlation between the combined variables is maximized. In practice, CCA has been mainly implemented as a substitute for univariate general linear model to link different modalities, and therefore, is a major and powerful tool in multimodal data fusion. However, the complicated multivariate formulations and obscure capabilities remain obstacles for CCA and its variants to being widely applied. We separately tested brain-behavioral association for language and mental health. Language scores were acquired by reading and picture vocabulary tests in the NIH toolbox. Mental health scores included depression, anxiety, somatic, avoidant, ADHD, and antisocial problems, which were assessed by Adult Self-Report and DSM-Oriented Scale.

The Picture Vocabulary Test is a CAT format measure of general vocabulary knowledge for ages 3–85 and is considered to be a strong measure of crystallized abilities (those abilities that are more dependent upon past learning experiences and are consistent across the lifespan). The participant is presented with an audio recording of a word and four photographic images on the computer screen and is asked to select the picture that most closely matches the meaning of the word. Higher scores indicate higher vocabulary ability. The Reading Test is a CAT format measure of reading decoding skill and of crystallized abilities, those abilities that are generally more dependent upon past learning experiences and consistent across the life span for ages 7-85. The participant is asked to read and pronounce letters and words as accurately as possible. Higher scores indicate better reading ability. Age-adjusted Scale Score: Participant

score is normed using the age-appropriate band of Toolbox Norming Sample (bands of ages 18-29, or 30-35), where a score of 100 indicates performance that was at the national average and a score of 115 or 85, indicates performance 1 SD above or below the national average for participants' age group. We used age-adjusted language scores in the current study.

The Adult Self-Report is a 126-item self-report questionnaire for adults (ages 18–59) assessing aspects of adaptive functioning and problems. The questionnaire provides scores for the following syndrome scales: anxious/depressed, withdrawn, somatic complaints, thought problems, attention problems, aggressive behavior, rule-breaking behavior, and intrusive behavior. The questionnaire provides scores for the following DSM-oriented scales: depressive problems, anxiety problems, somatic problems, avoidant personality problems, attention deficit/ hyperactivity problems (inattention and hyperactivity/impulsivity subscales), and antisocial personality problems. Additionally, the questionnaire asks about use of the following substances: tobacco, alcohol, and drugs. Items are rated on a 3-point scale: 0-Not True, 1-Somewhat or Sometimes True, 2-Very True or Often True. We used DSM-oriented scale and gender and age-adjusted T-scores as mental health scores in the current study.

We also used pseudo-randomised resampling (clustering the same number of twins for each sample) to test the robustness of brain-behavior association and the brain loading patterns. We started by withdrawing 10% to 90% data and iterated these steps 100 times. Therefore, for each percentage data withdrawal, there would be 100 CCA models and correlations to the anterior-posterior asymmetry pattern. We showed the charts with mean and standard error bars for each percentage data withdrawal.

**Reporting summary**
Further information on research design is available in the Nature Portfolio Reporting Summary linked to this article.

## Data availability
The neuroimaging data we used in the current study are available through the holders' website, which have been mentioned in the Methods Datasets section. Source data are provided. Source data are provided with this paper.

## Code availability
All the scripts and visualization are openly available at a GitHub repository (https://github.com/wanb-psych/microstructural_asymmetry). The packages are completely open for use, see documentations: BigBrainWarp (https://bigbrainwarp.readthedocs.io/en/latest/), BrainStat (https://brainstat.readthedocs.io/en/), BrainSpace (https://brainspace.readthedocs.io/en/latest/), SOLAR, and Scikit-learn (https://scikit-learn.org/stable/).

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

## Acknowledgements

We are grateful to those open datasets including BigBrain, HCP, NSPN, and MICs. BCB acknowledges research support from the National Science and Engineering Research Council of Canada (NSERC Discovery-1304413), Canadian Institutes of Health Research (FDN-154298, PJT-174995, PJT-191853), SickKids Foundation (NI17-039), BrainCanada, FRQ-S, the Tier-2 Canada Research Chairs program, and Healthy Brains, Healthy Lives. SLV and BCB are furthermore funded by the Helmholtz International BigBrain Analytics and Learning Laboratory (HIBALL), supported by the Helmholtz Association's Initiative and Networking Fund and the Healthy Brains, Healthy Lives initiative at McGill University. MDH is funded by the German Ministry for Education and Research (BMBF) and the Max Planck Society. SLV is supported by the Otto Hahn Award at Max Planck Society, and BW is supported by the International Max Planck Research School on Neuroscience of Communication: Function, Structure, and Plasticity (IMPRS NeuroCom), Graduate Academy Leipzig, and Mitacs Globalink Research Award.

## Author contributions

Conceptualization: B.W., S.L.V. Methodology: B.W., A.S., H.L.S., S.L.V. Formal analysis: B.W. Writing - Original Draft: B.W., S.L.V. Writing - Review & Editing: B.W., A.S., C.P., H.L.S., M.D.H., J.R., A.J., L.D., Ş.B., R.A.I.B., S.B.E., B.C.B., S.L.V. Visualization: B.W. Data curation: B.W., A.S., C.P., M.D.H., J.R., L.D., B.C.B., S.L.V. Project administration: B.W. Funding acquisition: B.W., B.C.B., S.L.V. Supervision: B.C.B., S.L.V.

## Funding

## Competing interests

The authors declare no competing interests.
