## [Transparent Peer Review file · Nature Communications]

Microstructural asymmetry in the human cortex

Corresponding Author: Mr Bin Wan

Version 0:

Reviewer comments:

Reviewer #1

(Remarks to the Author)

This is a very interesting study that uses previously-generated datasets to investigate microstructural asymmetry in the human brain in a novel way, and also addresses the possible behavioral relevance. The analyses seem generally sound and the findings well supported by the available data, although the data have limitations and I have some clarification questions on the methods.

For transparency the abstract could make clear which datasets, data/measure types and sample sizes were used for the different steps, including mention of a single post mortem brain for the first part, and T1w/T2w and other metrics for later steps. These points also relate to the limitations of available data and types for probing cortical microstructure, that can be discussed more fully (N of 1, indirect measures, different measures at different steps).

From the methods section 'Asymmetry Index': 'For HCP, MICs, and NSPN, we obtained the mean cortical intensity map, then z-scored the map for left and right hemispheres separately.' From this description it is not clear to me how subject-specific asymmetry maps were calculated in these datasets. In particular I am unsure if subject-specific asymmetry indexes were partly confounded with bilateral magnitudes that might relate to global measures such as average cortical thickness over the whole brain, or brain size. This in turn might have inflated the heritability values and associations with behavioral scores, as we know for example that brain size is quite strongly heritable and associated with behavioral scores.

From the Discussion: 'Previous work has suggested that the anterior-posterior asymmetry pattern in T1w/T2w is partly generated by the transmit field rather than the microstructure itself 8'. This seems a crucial point and readers without imaging backgrounds would benefit from further explanation. How, and to what extent, does the transmit field create an anterior-posterior asymmetry pattern? What does this mean for the interpretation of the study?

Did the heritability analysis allow for a shared environmental component of variance in the twins? If not, the heritability values may have been inflated.

I find figure 1F difficult to interpret. Are the different networks spatially overlapping? If so, a different visualization may be needed. Related to this, the sentence in the Results that refers to this figure panel could briefly indicate how skewness was calculated, or at least what it means conceptually, to help the reader along. I did not find skew mentioned in the Methods where a full explanation is needed.

From the Results: 'Our analysis focused on intra-hemispheric (i.e., LH_LH and RH_RH) connectivity'. Given previous literature and hypotheses, it would be of interest to know whether higher inter-hemispheric connectivity was associated with lower asymmetry, and the authors might already have the measures to assess this.

I think it would be better not to refer to 'replication' when the microstructural measure was not the same across the HCP, MICs, and NSPN datasets. Showing concordant evidence from different measures in different datasets is a strength, but if replication was a goal, then perhaps other datasets with the same measure as HCP could be used.

Reviewer #2

(Remarks to the Author)

Review of NCOMMS-24-23589-T

Classically, structural hemispheric asymmetries in the human brain get classified into one of three categories: Macrostructural asymmetries, microstructural asymmetries and molecular asymmetries. The vast majority of published research is focusing on macrostructural asymmetries, given the relative ease to assess them in-vivo in the human brain using neuroimaging methods such as MRI, etc.

Microstructural asymmetries, in comparison, have rarely been the focus of research in the past (but some previous studies

exist). Thus, the manuscript "Microstructural asymmetry in the human cortex" by Bin Wan and co-workers clearly fills a gap in the literature and has the potential to be published in a prestigious journal like NCOMMS. I do not see any major issues that would prevent publication that cannot be remedied by standard revision, but I do have some suggestions that the authors may wish to consider for a revised version of their work.

Specific comments:

Abstract:

Please describe the mentioned anterior-posterior pattern in more detail. Was there a leftward or a rightward asymmetry anterior / posterior and in which layers? This is the key take-away from the paper and it is presently unclear.

Introduction:

The authors could update their literature search on relevant papers on microstructural asymmetries for the introduction, I think some relevant works were missing (maybe because the term asymmetry is not always in title), e.g.:

Zachlod, D., Kedo, O., & Amunts, K. (2022). Anatomy of the temporal lobe: From macro to micro. *Handbook of clinical neurology*, 187, 17–51. <https://doi.org/10.1016/B978-0-12-823493-8.00009-2>

Chance S. A. (2014). The cortical microstructural basis of lateralized cognition: a review. *Frontiers in psychology*, 5, 820. <https://doi.org/10.3389/fpsyg.2014.00820>

Kedo, O., Zilles, K., Palomero-Gallagher, N., Schleicher, A., Mohlberg, H., Bludau, S., & Amunts, K. (2018). Receptor-driven, multimodal mapping of the human amygdala. *Brain structure & function*, 223(4), 1637–1666. <https://doi.org/10.1007/s00429-017-1577-x>

Also, in general I would suggest to include a few more sentences on which specific results have been published for microstructural asymmetries and for which dependent variables. Just writing "there is limited evidence" is not enough, this evidence should be described.

Did the authors have any hypotheses for their project based on the published literature or was the project purely data-driven? If hypotheses existed, these should be mentioned.

Methods:

For all datasets used, the handedness of the tested participant should be indicated. Surely, this must be known for BigBrain and the other used datasets. Handedness is associated with structural asymmetries at the macroscale and thus it could potentially also influence asymmetries at the microscale:

Sha, Z., Pepe, A., Schijven, D., Carrión-Castillo, A., Roe, J. M., Westerhausen, R., Joliot, M., Fisher, S. E., Crivello, F., & Francks, C. (2021). Handedness and its genetic influences are associated with structural asymmetries of the cerebral cortex in 31,864 individuals. *Proceedings of the National Academy of Sciences of the United States of America*, 118(47), e2113095118. <https://doi.org/10.1073/pnas.2113095118>

If this is not known, it should be discussed as a limitation. Especially for the n=1 BigBrain study this may be highly relevant. For the HCP dataset, handedness should be available, for MIC and NSPN also.

Asymmetry index:

It was not clear to me why three different ways to calculate the AI were used. Most studies use $(RH-LH)/(LH+RH)$ and it intuitively would make sense to use the same calculation for all datasets. Using a subtraction $LH-RH$ for BigBrain makes it difficult to compare these data to other studies. I would suggest the authors streamline their AI approach, or explain in more detail why this approach with three different AIs was used.

Effects of sex and age

Effects of handedness need to be included here, too.

Brain-behavioral association

Again, handedness as one of the most relevant behaviors for structural asymmetries needs to be included here.

Results

The authors rely heavily on figures to communicate their results. This is fine for the manuscript, but makes later integration into meta-analyses highly difficult to impossible. While the data is available, running the scripts etc to obtain the results again would also be really cumbersome for any scientists interested in including the data from this study in a future meta-analysis. I would therefore strongly encourage the authors to supplement their figures in the results with supplementary tables stating the exact AI values for the different analysis and also the LH / RH values and effect sizes. This would strongly increase the value of the study for future meta-analyses.

Please include handedness data whenever available in the results section.

Discussion

The statement "Asymmetry in structural and functional brain organization is implicated in key human cognitive functions, including language, and is associated with neuropsychiatric conditions." needs a reference or two, e.g.:
Hartwigsen, G., Bengio, Y., & Bzdok, D. (2021). How does hemispheric specialization contribute to human-defining cognition?. *Neuron*, 109(13), 2075–2090. <https://doi.org/10.1016/j.neuron.2021.04.024>
Ocklenburg & Güntürkün (2024). *The Lateralized Brain. The Neuroscience and Evolution of Hemispheric Asymmetries*. Academic Press.

Limitations:

Of course, the authors should clearly state that potential issues of an $n=1$ study for a bimodal trait like asymmetry. I was surprised to not see any statement on this issue in the discussion.

Also potential effects of handedness need to be discussed in this context.

Reference list:

Please check the reference list, some references do not fit journal format and are for example written in all-caps.

Reviewer #3

(Remarks to the Author)

Wan and colleagues presented a massive analysis of the human cortex asymmetry based on BigBrain one subject data, HCP data of 1206 subjects, MICs data of 50 subjects, and NSPN data of 2245 subjects.

First of all, the presented work consists a lot of different types of analyses in order to extract grey matter asymmetry patterns and localise a similarity/repeatability of the revealed patterns over the difference samples. However, all presented findings have no one common motivation allowing one to read the manuscript as a self consistent story. Authors used one subject analysis from BigBrain project as a tip for the similarity between cytoarchitectural and in vivo asymmetry. I believe, it is not too convincing point here. I would expect the opposite situation, when in vivo based group analysis allows to reproduce the similar patterns in BigBrain data at layers' scale. Moreover, in all cases the "similarity" should have a qualitative expression for spatial patterns.

In turn, it is not clear how did authors connected layerwise geometry and stained intensities with either T1w/T2w ratio or qT1/MT measures. Physically, these measures could reflect quite different microstructural features, not necessarily coinciding, in particular, for such a proxy of myelination as T1w/T2w ratio. It is unclear, how are correlated T1w/T2w derived patterns with qT1/MT? Which provides a better explanation of variance change?

What happens if these parameters would be divided into sub-thickness with different thickness in accordance with FreeSurfer decomposition? If we keep in mind that grey matter thickness usually alters between 2-4 mm, then sub-surfaces at 1mm scale might affect the correlations. Would it affect the found correlations in Fig 2C as well in order to prove or reject it?

There are some minor remarks:

it is unclear for me why authors used different AI notations? Would it be better to use one for all, for example, the following from Ref.

<https://academic.oup.com/mbe/article/40/9/msad181/7240668> as $\arcsin()$ function or could be some similar form as \arctan .

Colour coding in Fig 1 is really challenging to understand over the whole figure.

it is unclear meaning in Fig 1C: mean intensity map and residuals. What exactly this figure does?

in Fig 2C, for the colour combination of yellow-white, it is almost impossible to see the correlations

I did not find any tables in the supplementary.

Often the authors referred to significant correlations with p-values, however, it is more informative to see in parallel the effect size.

I have read a few times what authors would like to say in Fig. 3 and it was too entangled in both in the text and figure captions. Authors should elaborate it in order to give a more clear message.

It is much better to find a number for "similarity" every time it is used in the text. In particular, when it concerns spatial patterns including differences in layers.

Version 1:

Reviewer comments:

Reviewer #1

(Remarks to the Author)

Many thanks to the authors for addressing my points.

I think the issue of heritability calculation still remains. The current analysis does not separate genetic influences from shared environmental influences within families. Twin studies are designed to separate these (by applying a standard ACE model rather than an AE model, where C is the shared family environment component). In the rebuttal letter, the authors say that Kochunov et al. (2019) found the AE model to be more accurate for estimating heritability than ACE in the HCP dataset. This finding was not obvious to me in Kochunov's results or recommendations (different analysis methods were applied so that the specific effect of ACE vs AE models may not have been clear). Also the brain measures in the present study are not the same as studied by Kochunov et al.. Regardless of the findings by Kochunov et al., an ACE model would better be tested as a sanity check on the AE results in the present study. Otherwise the reader is left wondering whether there is significant heritability at all, or whether the results reflect a familial effect of shared environment instead.

(By the way, the revised text on heritability in the revised Methods section of the manuscript (page 23) cites reference number 79, but in the reference list 79 is a different paper, and Kochunov et al. does not appear. Please check citation-reference matching.)

Thank you for describing what the skew measure indicates in the revised Results text. I think the Methods section still misses the exact formulation of this measure.

Reviewer #2

(Remarks to the Author)

The authors have improved an already strong manuscript and I can recommend acceptance now.

Reviewer #3

(Remarks to the Author)

I have no additional comments to Authors. All my concerns were attended in details with a lot of information to clarify each point.

NCOMMS-24-23589-T

Response Letter:

We would like to thank the Editor(s) and Reviewers for their positive evaluations, constructive comments, and for the opportunity to submit a revised manuscript. We feel that the comments and suggestions have greatly improved our manuscript. Please find our detailed responses below according to the Reviewers' comments one by one. For the revised manuscript, changes are highlighted in yellow and are also listed in the response. Github scripts have been updated and source data are provided with re-submission. Acknowledgements, author contributions, and competing interests have been moved to after the references section. Regarding the figures in the main text, we also provide additional pdf (vector figures) for re-editing for our re-submission.

Reviewer #1:

“This is a very interesting study that uses previously-generated datasets to investigate microstructural asymmetry in the human brain in a novel way, and also addresses the possible behavioral relevance. The analyses seem generally sound and the findings well supported by the available data, although the data have limitations and I have some clarification questions on the methods.”

We thank the Reviewer for their appreciation of our work and the helpful comments. We have detailed answers to the questions and comments below.

Q1: “For transparency the abstract could make clear which datasets, data/measure types and sample sizes were used for the different steps, including mention of a single post mortem brain for the first part, and T1w/T2w and other metrics for later steps. These points also relate to the limitations of available data and types for probing cortical microstructure, that can be discussed more fully (N of 1, indirect measures, different measures at different steps).”

A1: We have now revised the abstract. Please see below:

“While macrostructural brain asymmetry and its relevance to human cognitive function have been consistently shown, the underlying neurobiological signatures remain an open question. Here, we probe layer-specific microstructural asymmetry of the human cortex using intensity from post-mortem cytoarchitecture (BigBrain, N = 1). Overall, anterior and posterior regions show leftward and rightward asymmetry respectively, but this pattern varied across cortical layers. A similar anterior-posterior pattern is observed

using in vivo Human Connectome Project (HCP, N = 1101) T1w/T2w microstructure, with asymmetry showing the strongest similarity with post-mortem-based asymmetry of layer III. Moreover, microstructural asymmetry varied as a function of age and sex, and was heritable. Asymmetry in microstructure corresponded to asymmetry of intrinsic function, particularly in sensory areas. Probing the behavioral relevance, we observed differential association of language and markers of mental health with asymmetry, illustrating a functional divergence between inferior-superior and anterior-posterior microstructural axes possibly anchored in development. Last, we showed concordant evidence of cortical asymmetry with different in vivo microstructural measures including magnetization transfer (MT, N = 286) and quantitative T1 (qT1, N = 50). Together, our study highlights the layer-based patterning of post-mortem microstructural asymmetry of the human cortex and uses an in vivo model to illustrate how microstructural asymmetry is associated with intrinsic function and behavior”

Q2: “From the methods section ‘Asymmetry Index’: ‘For HCP, MICs, and NSPN, we obtained the mean cortical intensity map, then z-scored the map for left and right hemispheres separately.’ From this description it is not clear to me how subject-specific asymmetry maps were calculated in these datasets. In particular I am unsure if subject-specific asymmetry indexes were partly confounded with bilateral magnitudes that might relate to global measures such as average cortical thickness over the whole brain, or brain size. This in turn might have inflated the heritability values and associations with behavioral scores, as we know for example that brain size is quite strongly heritable and associated with behavioral scores.”

A2: We are happy to further clarify. To account for potential bias from bilateral hemispheric differences (global measure), we z-scored the intensity score for left and right hemispheres separately for each subject, and the following obtained the asymmetry index. Z-scoring makes both left and right hemisphere mean equal to zero. We also calculated heritability and behavioral association based on this asymmetry index. However, when using non-z-scored (raw intensity) asymmetry index, as suggested by other Reviewers $(LH - RH)/(LH + RH)$, our results remained largely unchanged, despite a subtle change in asymmetry in medial prefrontal regions. We show the figure below for comparison and added this information as a supplementary figure and in our Results section:

“To further test the robustness of our findings, we also used a raw intensity score to calculate the asymmetry index by $(LH - RH)/(LH + RH)$. Overall findings were consistent but showed a subtle difference (all spatial Pearson $r > 0.9$) in the medial frontal cortex for asymmetry, sex and age t-maps (Supplementary Figure S7).”

Supplementary Figure S7. Raw score versus z-score for asymmetry index.

We also showed the average cortical thickness effect, please see below. About 10% of parcels statistically survived multiple comparison correction. Because we already considered averaged intensity (z-score above), we would not include average cortical thickness in our study.

Mean cortical thickness effects

In addition, we now consider the regional cortical thickness. We calculated the index intensity by cortical thickness and compared with the original results, it was highly similar ($r = 0.996$) see the below figure (Supplementary Figure S8).

Supplementary Figure S8. Asymmetry of microstructural intensity by cortical thickness.

Q3: “From the Discussion: ‘Previous work has suggested that the anterior-posterior asymmetry pattern in T1w/T2w is partly generated by the transmit field rather than the microstructure itself 8’. This seems a crucial point and readers without imaging backgrounds would benefit from further explanation. How, and to what extent, does the transmit field create an anterior-posterior asymmetry pattern? What does this mean for the interpretation of the study?”

A3: In the Human Connectome Project (HCP), the group-averaged transmit field (B1+ bias, also known as the Actual Flip Angle Imaging or AFI map) creates an anterior-posterior asymmetry pattern as described by (Glasser et al., 2022). However, using the AFI map to correct the transmit field may result in overcorrection as B1+ is sensitive to the structural imaging sequence, and HCP MP2RAGE T1w and T2w images have already been partially corrected for B1+ bias (Glasser et al., 2013; Glasser & Essen, 2011).

To further test the robustness of our observations of left-right asymmetry patterns in HCP, as well as age and sex effects, we used two additional sequences/datasets. The first is the MP2RAGE sequence (qT1 image, MICA-MICs dataset), which is known to produce more accurate qT1 maps with reduced sensitivity to B1 inhomogeneities (Haast et al., 2016; Marques et al., 2010). The second is the multiparametric mapping sequence (including multiple multi-echo T1 images and MT images, NSPN dataset), where the algorithm (Weiskopf et al., 2013) corrects for B1+ bias in MT images. Both datasets showed similar asymmetry patterns to those found in HCP.

We have revised our Discussion to clarify these points. See below:

“Previous work has suggested that the anterior-posterior asymmetry pattern in T1w/T2w is partly generated by the transmit field rather than the microstructure itself⁸. However, T1w/T2w images have been corrected for some of the B1+ bias (see Methods), and using flip angles map to further correct the image might reduce the real signals⁸. In addition, we used qT1 and MT to validate our in vivo results and observed concordance across all metrics.”

Limitations:

“Although T1w/T2w images reveal a strong anterior-posterior asymmetry pattern and concordant validation from other microstructural measures have been found, it still requires more B1+ bias correction to reduce the inhomogeneities in HCP T1w/T2w.”

Q4: “Did the heritability analysis allow for a shared environmental component of variance in the twins? If not, the heritability values may have been inflated.”

A4: In the twin heritability analysis, we used A+E as suggested by prior work (Kochunov et al., 2019), as the A+E model has a higher accuracy of estimating heritability than the A+E+C (common environment) model in HCP (Kochunov et al., 2019). We have revised in the Methods section, see below:

“In this study, we quantified the heritability of asymmetry of functional gradients using the A + E model as suggested by prior study, as the A+E model has a higher accuracy of estimating heritability than the A+E+C (common environment) model in HCP⁷⁹.”

Q5: “I find figure 1F difficult to interpret. Are the different networks spatially overlapping? If so, a different visualization may be needed. Related to this, the sentence in the Results that refers to this figure panel could briefly indicate how skewness was calculated, or at least what it means conceptually, to help the reader along. I did not find skew mentioned in the Methods where a full explanation is needed.”

A5: Thanks for the suggestion. We have reorganized Figure 1, see below.

Skewness indicates the difference in intensity as a function of cortical depth. A higher skewness indicates that deep surfaces have a higher intensity relative to superficial surfaces, whereas a lower skewness indicates a less steep difference between upper and lower depths. A high left-right asymmetry of skewness indicates that the intensity distribution is more skewed on the left relatively deeper surfaces. Skewness of left-right asymmetry tells us the distribution of asymmetry across the layers with a high skewness score indicating that left-right asymmetry shifts on the deeper layers. For example, in the language network, upper layers show leftward asymmetry, while deeper layers show rightward asymmetry. A skewness score of 0.3 indicates that the left-right shift for language occurs in the middle-deeper layers. We have added this text in the last of Figure 1 results. See below:

“Skewness overall indicates the difference in intensity as a function of cortical depth. A higher skewness indicates that deep surfaces have a higher intensity relative to superficial surfaces, whereas a lower skewness indicates a less steep difference between upper and lower layers. A high left-right asymmetry of skewness indicates that the

intensity distribution is more skewed on the left relatively deeper surfaces. Skewness of left-right asymmetry tells us the distribution of asymmetry across the layers with a high skewness score indicating that left-right asymmetry shifts on the deeper layers.”

Q6: “From the Results: ‘Our analysis focused on intra-hemispheric (i.e., LH_LH and RH_RH) connectivity’. Given previous literature and hypotheses, it would be of interest to know whether higher inter-hemispheric connectivity was associated with lower asymmetry, and the authors might already have the measures to assess this.”

A6: Thanks for the good question. Intra- and inter-hemispheric functional connectivity matrices are highly similar, as observed in many papers using HCP data (Quinn et al., 2024; Raemaekers et al., 2018; Wan et al., 2022). We observed that higher inter-hemispheric functional connectivity is associated with higher leftward microstructural asymmetry (not lower asymmetry), see the figure below. However, inter-hemispheric functional connectivity asymmetry can not have a clear asymmetric orientation. For the bins that are purple, the symmetric bins are all green (in principle they should share the same score but not the opposite score).

We believe answering this question is out of scope in the current study of functional and microstructural asymmetry. It would be good to do a specific investigation of this question (inter-hemispheric functional connectivity) in the future for a new study.

Q7: “I think it would be better not to refer to ‘replication’ when the microstructural measure was not the same across the HCP, MICs, and NSPN datasets. Showing concordant evidence from different measures in different datasets is a strength, but if replication was a goal, then perhaps other datasets with the same measure as HCP could be used.”

A7: Thanks for the point. Indeed, the aim was to find concordant evidence with different microstructural measures. We have replaced replication with the term: *concordant validation*.

Reviewer #2:

“Classically, structural hemispheric asymmetries in the human brain get classified into one of three categories: Macrostructural asymmetries, microstructural asymmetries and molecular asymmetries. The vast majority of published research is focusing on macrostructural asymmetries, given the relative ease to assess them in-vivo in the human brain using neuroimaging methods such as MRI, etc. Microstructural asymmetries, in comparison, have rarely been the focus of research in the past (but some previous studies exist). Thus, the manuscript “Microstructural asymmetry in the human cortex” by Bin Wan and co-workers clearly fills a gap in the literature and has the potential to be published in a prestigious journal like NCOMMS. I do not see any major issues that would prevent publication that cannot be remedied by standard revision, but I do have some suggestions that the authors may wish to consider for a revised version of their work.”

We appreciate the Reviewer's positive feedback and constructive comments. Detailed responses to the questions and comments are provided below.

Q1: “Abstract

Please describe the mentioned anterior-posterior pattern in more detail. Was there a leftward or a rightward asymmetry anterior / posterior and in which layers? This is the key take-away from the paper and it is presently unclear. ”

A1: Thanks for the point. We have rewritten the abstract with more details, please see below:

“While macrostructural brain asymmetry and its relevance to human cognitive function have been consistently shown, the underlying neurobiological signatures remain an open question. Here, we probe layer-specific microstructural asymmetry of the human cortex using intensity from post-mortem cytoarchitecture (BigBrain, N = 1). Overall, anterior and posterior regions show leftward and rightward asymmetry respectively, but this pattern varied across cortical layers. A similar anterior-posterior pattern is observed using in vivo Human Connectome Project (HCP, N = 1101) T1w/T2w microstructure, with asymmetry showing the strongest similarity with post-mortem-based asymmetry of layer III. Moreover, microstructural asymmetry varied as a function of age and sex, and was heritable. Asymmetry in microstructure corresponded to asymmetry of intrinsic function, particularly in sensory areas. Probing the behavioral relevance, we observed differential association of language and markers of mental health with asymmetry, illustrating a functional divergence between inferior-superior and anterior-posterior microstructural axes possibly anchored in development. Last, we showed concordant evidence of cortical asymmetry with different in vivo microstructural measures including magnetization transfer (MT, N = 286) and quantitative T1 (qT1, N = 50). Together, our

study highlights the layer-based patterning of post-mortem microstructural asymmetry of the human cortex and uses an in vivo model to illustrate how microstructural asymmetry is associated with intrinsic function and behavior”

Q2: “Introduction

The authors could update their literature search on relevant papers on microstructural asymmetries for the introduction, I think some relevant works were missing (maybe because the term asymmetry is not always in title), e.g.:

Zachlod, D., Kedo, O., & Amunts, K. (2022). Anatomy of the temporal lobe: From macro to micro. Handbook of clinical neurology, 187, 17–51. <https://doi.org/10.1016/B978-0-12-823493-8.00009-2>

Chance S. A. (2014). The cortical microstructural basis of lateralized cognition: a review. Frontiers in psychology, 5, 820. <https://doi.org/10.3389/fpsyg.2014.00820>

Kedo, O., Zilles, K., Palomero-Gallagher, N., Schleicher, A., Mohlberg, H., Bludau, S., & Amunts, K. (2018). Receptor-driven, multimodal mapping of the human amygdala. Brain structure & function, 223(4), 1637–1666. <https://doi.org/10.1007/s00429-017-1577-x>”

Also, in general I would suggest to include a few more sentences on which specific results have been published for microstructural asymmetries and for which dependent variables. Just writing “there is limited evidence” is not enough, this evidence should be described.”

A2: We now added mentioned and more references. See below:

“There are a few studies focusing on microstructural asymmetry in language regions ^{7,21–24} and amygdala ²⁵. For example, microstructural intensity in language areas (BA areas 44 and 45) is higher in the left hemisphere ⁷ and left-right differences in amygdala subnuclear volumes measured by cytoarchitectural mapping ²³.”

Q3: “Introduction

Did the authors have any hypotheses for their project based on the published literature or was the project purely data-driven? If hypotheses existed, these should be mentioned.”

A3: Thanks for the point. We have now added our main aims and expectations more explicitly in the introduction. Please also see below:

“Motivated by previous work showing cortex-wide patterns of asymmetry in macrostructural markers such as cortical thickness and surface area ^{10,12,14}, and regional

reports of asymmetry in cortical microstructure^{7,21-25}, we aimed to study microstructural asymmetry across the whole cortex.”

*“We furthermore aimed to probe its functional relevance, motivated by the Structural Model, stating that microstructural similarity relates to connectivity*¹⁶, and previous work on the functional markers of asymmetry^{6,13,14}.”

Q4: “Methods

For all datasets used, the handedness of the tested participant should be indicated. Surely, this must be known for BigBrain and the other used datasets. Handedness is associated with structural asymmetries at the macroscale and thus it could potentially also influence asymmetries at the microscale.

Sha, Z., Pepe, A., Schijven, D., Carrión-Castillo, A., Roe, J. M., Westerhausen, R., Joliot, M., Fisher, S. E., Crivello, F., & Francks, C. (2021). Handedness and its genetic influences are associated with structural asymmetries of the cerebral cortex in 31,864 individuals. *Proceedings of the National Academy of Sciences of the United States of America*, 118(47), e2113095118. <https://doi.org/10.1073/pnas.2113095118>

If this is not known, it should be discussed as a limitation. Especially for the n=1 BigBrain study this may be highly relevant.”

A4: Thanks for the point. Unfortunately, handedness for the BigBrain subject is unknown. Based on your question, we tested the effects of handedness in HCP. We found, when controlling for sex and age, no regions to survive multiple comparisons. See below:

Results:

“In addition, we tested for potential association between handedness and microstructural asymmetry. We found no parcels that survived statistical thresholds.”

Discussion:

“Of note, in the current work, we found no association between handedness and microstructural asymmetry. Other work either reports little association between handedness and cortical thickness and surface area in the multi-center ENIGMA data¹⁰. However, handedness and polygenic risk scores of handedness are associated with macrostructural asymmetry in a few regions in the UK Biobank⁷⁶. Future studies may focus on meta-analysis to identify whether handedness is associated with microstructural asymmetry with more papers published or use more fine grained investigations of dexterity and brain anatomy.”

Q5: “Methods**For the HCP dataset, handedness should be available, for MIC and NSPN also.”**

A5: Thank you for noting this. Please see our above answer on handedness effects on asymmetry in our main study sample. Based on this we refrained from testing handedness associations in the other datasets. However, we now also include considerations with respect to bifurcations and handedness in our discussion, please see below:

“Of note, in the current work, we found no association between handedness and microstructural asymmetry. Other work either reports little association between handedness and cortical thickness and surface area in the multi-center ENIGMA data¹⁰. However, handedness and polygenic risk scores of handedness are associated with macrostructural asymmetry in a few regions in the UK Biobank⁷⁶.”

Q6: “Methods Asymmetry index**It was not clear to me why three different ways to calculate the AI were used. Most studies use (RH-LH)/(LH+RH) and it intuitively would make sense to use the same calculation for all datasets. Using a subtraction LH-RH for BigBrain makes it difficult to compare these data to other studies. I would suggest the authors streamline their AI approach, or explain in more detail why this approach with three different AIs was used.”**

A6: The BigBrain data is a 60x360 matrix (layer by region). It is recommended to correct the mean intensity score along the layers (often referred to as intracortical depth) before using it (Paquola, Wael, et al., 2019). Therefore, we obtained the left and right standardized residuals separately for the big brain and then calculated the asymmetry as LH - RH.

For the *in vivo* data sets, each subject had a 1x360 matrix. We z-scored the left and right hemispheres separately before calculating the asymmetry as LH - RH. This approach is preferable to (LH-RH)/(LH+RH) because z-scoring theoretically overcomes the problem of unequal bilateral mean intensity (where the sum of left and right is not equal). Normalizing asymmetry using (LH+RH) may theoretically increase the measured asymmetry in certain cases, such as in region A: LH = 0.1, RH = -0.1 vs. region B: LH = 0.2, RH = 0. This would thus make further normalization using LH+RH unnecessary. We also compared two approaches in Supplementary Figure S7, please see below:

Supplementary Figure S7. Raw score versus z-score for asymmetry index.

Conversely, we used $(LH-RH)/(LH+RH)$ for the functional connectivity matrix. Here all values are positive, and the connectivity matrices are Fisher Z-transformed and we account for any possible local variability in intensity to get at a relative measure of asymmetry. However, results were similar irrespective of whether we used $(LH-RH)/(LH+RH)$ or $LH-RH$ to compute functional asymmetry. Please find the comparison Supplementary Figure S9 below:

Supplementary Figure S9. Comparisons between original asymmetry $(LH-RH)/(LH+RH)$ calculation and $LH-RH$ for functional data.

Q7: “Methods

Effects of sex and age

Effects of handedness need to be included here, too.

Brain-behavioral association, Again, handedness as one of the most relevant behaviors for structural asymmetries needs to be included here.”

A7: Please see our answer to Q5-6 on handedness effects on asymmetry. We found no statistical association between handedness and asymmetry.

Q8: “Methods

The authors rely heavily on figures to communicate their results. This is fine for the manuscript, but makes later integration into meta-analyses highly difficult to impossible. While the data is available, running the scripts etc to obtain the results again would also be really cumbersome for any scientists interested in including the data from this study in a future meta-analysis. I would therefore strongly encourage the authors to supplement their figures in the results with supplementary tables stating the exact AI values for the different analysis and also the LH / RH values and effect sizes. This would strongly increase the value of the study for future meta-analyses.”

A8: We agree, that is an important point! Now we provide the Source Data with resubmission, where tables including LH and RH score, asymmetry, effect sizes are reported.

Q9: “Methods

Please include handedness data whenever available in the results section.”

A9: Please see our above answer to Q5-6 on handedness effects on asymmetry. We have also added the number of subjects with handedness: 1002 subjects had a handedness score ≥ 0 and 99 subjects had a handedness score < 0 , and mean (SD) handedness score is 66.0 (44.0).

Q10: “Discussion

The statement “Asymmetry in structural and functional brain organization is implicated in key human cognitive functions, including language, and is associated with neuropsychiatric conditions.” needs a reference or two, e.g.:

Hartwigsen, G., Bengio, Y., & Bzdok, D. (2021). How does hemispheric specialization contribute to human-defining cognition?. *Neuron*, 109(13), 2075–2090. <https://doi.org/10.1016/j.neuron.2021.04.024>

Ocklenburg & Güntürkün (2024). *The Lateralized Brain. The Neuroscience and Evolution of Hemispheric Asymmetries*. Academic Press.”

A10: Thanks for spotting this omission, we have now added the relevant references!

Q11: “Limitations:

Of course, the authors should clearly state that potential issues of an n=1 study for a bimodal trait like asymmetry. I was surprised to not see any statement on this issue in the discussion.”

A11: We agree and have underscored this limitation. Please see the last second paragraph in the discussion for details. You can also see it here:

“Firstly, though the BigBrain (N = 1) offers unique insights into cortical microstructure at ultra high resolution, and links to our in vivo model, the results are limited to one subject. Further work on ultra-high resolution neuroimaging (e.g., 7T or 9.4T MRI), sensitive to laminar changes, will aid in also understanding layer-level markers of individual variation.”

Q12: “Also potential effects of handedness need to be discussed in this context.”

A12: Please see our above answer on handedness effects on asymmetry. We have added this information to the results and discussion.

Q13: “Reference list

Please check the reference list, some references to not fit journal format and are for example written in all-caps.”

A13: Thanks! We now have re-edited the references and the journal formatting editor will also help work on it.

Reviewer #3:

“Wan and colleagues presented a massive analysis of the human cortex asymmetry based on BigBrain one subject data, HCP data of 1206 subjects, MICs data of 50 subjects, and NSPN data of 2245 subjects.”

We appreciate the Reviewer's helpful comments about our work. Below, we provide detailed responses to each question and comment.

Q1: “First of all, the presented work consists a lot of different types of analyses in order to extract grey matter asymmetry patterns and localise a similarity/repeatability of the revealed patterns over the difference samples. However, all presented findings have no one common motivation allowing one to read the manuscript as a self consistent story. Authors used one subject analysis from BigBrain project as a tip for the similarity between cytoarchitectural and in vivo asymmetry. I believe, it is not too convincing point here. I would expect the opposite situation, when in vivo based group analysis allows to reproduce the similar patterns in BigBrain data at layers' scale. Moreover, in all cases the "similarity" should have a qualitative expression for spatial patterns. ”

A1: Thanks for the point. We apologize if our motivation may not have been clear in the introduction. In short, while macrostructural asymmetry has been widely studied, underlying micro- and meso-structure markers such as cytoarchitecture and myeloarchitecture of cortical regions have not been studied at the whole brain level. Yet microstructural measurements provide further insights in the neurobiological basis of macroscale structural asymmetry and help understand how structural asymmetry supports functional asymmetry in the human brain.

For cytoarchitecture, we used the BigBrain dataset, which provides a whole-brain 3D model of cell distribution (Amunts et al., 2013, 2020). This dataset represents a decade of work by hundreds of researchers to create a digital brain. Although it represents only one subject, BigBrain provides detailed information at the cellular level that exceeds what large in vivo datasets can provide.

We acknowledge the limitations of using a single subject in our discussion and noted that a second BigBrain dataset will be released soon. This will allow us to revisit and verify the robustness of the cytoarchitectural asymmetry findings. We have revised the relevant sections as shown below.

Introduction:

“While macrostructural asymmetry has been widely studied, underlying micro and mesostructure markers such as cytoarchitecture and myeloarchitecture of cortical regions have largely been studied in isolation.”

“Motivated by previous work showing cortex-wide patterns of asymmetry in macrostructural markers such as cortical thickness and surface area^{10,12,14}, and regional reports of asymmetry in cortical microstructure^{7,21-25}, we aimed to study microstructural asymmetry across the whole cortex.”

“We furthermore aimed to probe its functional relevance, motivated by the Structural Model, stating that microstructural similarity relates to connectivity¹⁶, and previous work on the functional markers of asymmetry^{6,13,14}.”

“Given that studies on macroscale structural asymmetry have reported leftward asymmetry in frontal regions and rightward asymmetry in occipital regions, we wished to study what the underlying microstructural correlates of these macrostructural patterns would be.”

Limitation:

“Firstly, though the BigBrain (N = 1) offers unique insights into cortical microstructure at ultra high resolution, and links to our in vivo model, the results are limited to one subject. Further work on ultra-high resolution neuroimaging (e.g., 7T or 9.4T MRI), sensitive to laminar changes, will aid in also understanding layer-level markers of individual variation”

Q2: “In turn, it is not clear how did authors connected layerwise geometry and stained intensities with either T1w/T2w ratio or qT1/MT measures. Physically, these measures could reflect quite different microstructural features, not necessarily coinciding, in particular, for such a proxy of myelination as T1w/T2w ratio. It is unclear, how are correlated T1w/T2w derived patterns with qT1/MT? Which provides a better explanation of variance change?”

A2: Thanks for the question. To clarify, the correlations we presented are spatial associations. We evaluated the significance of these correlations while accounting for the spatial autocorrelation inherent in the brain by creating surrogate null maps that preserve spatial autocorrelation assessed using variograms (Burt et al., 2020). The theory behind spatial autocorrelation is that features in brain regions are not independent and may be related to the neighboring or distant regions. The variogram technique uses the geometric matrix to generate brain scores that could plausibly arise from neighboring or distant regions (Burt et al., 2020; Vos de Wael et al., 2020). We performed 1000 permutations to generate surrogate maps and correlated them with the target

map to obtain a p-value indicating how often autocorrelated surrogate maps had correlations that exceeded those of the actual maps.

The geometric matrix we used is derived from the HCP group-averaged template. While we acknowledge that spatial correlations do not necessarily imply causal relationships, we provided correlation coefficients to indicate the degree of correlation between the two maps. Given that T1w/T2w, qT1, and MT all measure microstructure, we expect similar asymmetry patterns across measures. Indeed, related work has shown that these features generally show similar spatial distributions across the cortex (Paquola, Bethlehem, et al., 2019; Paquola, Wael, et al., 2019; Paquola & Hong, 2023).

For variance changes, this is a complex topic in neurophysics that cannot be fully addressed here due to its depth and complexity, requiring a more extensive discussion beyond the scope of this response. Although all MRI measurements are sensitive to myelination, they still have differences (Paquola & Hong, 2023). Myelin water fraction is based on the fact that in the central nervous system, the T2 signal decay follows a multi-exponential curve. By fitting the decay curve using least-square methods, these models can distinguish the myelin water pool, with ultrashort T2 decay, from the non-myelin water pool (T1w/T2w). T1w/T2w also includes non-neuronal signals. In contrast, magnetization transfer (MT) indirectly measures myelin based on the exchange and cross-relaxation between macromolecules (found predominantly in myelin) and water. Longitudinal relaxation rate (R1, also reported as 1/T1, quantitative T1) is similarly driven by the cross-relaxation of lipids and water. We have added this information in the Discussion, see below:

“In addition, although all MRI measurements are sensitive to myelination, they still have differences⁶⁶. Theoretically MT and qT1 detect exchange and cross-relaxation between lipids and water in tissue, and T1w/T2w means to correlate based on neurobiological principles by contrast of fitting the decay curve using least-square methods in non-myelin water pool⁶⁷. Compared to MT and qT1, T1w/T2w includes more non-myelin signals.”

Q3: “What happens if these parameters would be divided into sub-thickness with different thickness in accordance with FreeSurfer decomposition? If we keep in mind that grey matter thickness usually alters between 2-4 mm, then sub-surfaces at 1mm scale might affect the correlations. Would it affect the found correlations in Fig 2C as well in order to prove or reject it?”

A3: We studied equi-volumetric surfaces in case of the HCP data. We summarized them into four sub-surfaces as shown in the figure below. However, when we studied the asymmetry pattern for each surface, they were highly consistent with each other. We believe the in vivo imaging technique at present is sensitive to capture the intracortical variation, but regarding the subtle

asymmetry across cortical depth, it still requires a higher resolution, as we only observed subtle variations in asymmetry that had the same overall anterior-posterior pattern. Thus we only focused on the in vivo overall asymmetry in this manuscript. BigBrain can capture the intracortical asymmetry due to its ultra-high-resolution and offers insights that asymmetry may vary along layers. Therefore, we only correlate the overall asymmetry in HCP with BigBrain layer-specific to see which layer is more similar to in vivo.

Supplementary Figure S10. T1w/T2w asymmetry across four equivolumetric surfaces.

In addition, we calculated the index intensity by cortical thickness and compared with the original results, it was highly similar ($r = 0.996$) see the below figure.

Supplementary Figure S8. Asymmetry of microstructural intensity by cortical thickness.

Q4: “it is unclear for me why authors used different AI notations? Would it be better to use one for all, for example, the following from Ref. <https://academic.oup.com/mbc/article/40/9/msad181/7240668> as arcsin() function or could be some similar form as arctan.”

A4: We are happy to explain this decision. The BigBrain data is a 60x360 matrix (layer by region). It is recommended to correct the mean intensity score along the layers (often referred to as intracortical depth) before using it (Paquola, Wael, et al., 2019). Therefore, we obtained the left and right standardized residuals separately for the big brain and then calculated the asymmetry as LH - RH.

For the *in vivo* data sets, each subject had a 1x360 matrix. We z-scored the left and right hemispheres separately before calculating the asymmetry as LH - RH. This approach is preferable to (LH-RH)/(LH+RH) because z-scoring theoretically overcomes the problem of unequal bilateral mean intensity (where the sum of left and right is not equal). Normalizing asymmetry using (LH+RH) may theoretically increase the measured asymmetry in certain cases, such as in region A: LH = 0.1, RH = -0.1 vs. region B: LH = 0.2, RH = 0. This would thus make further normalization using LH+RH unnecessary. We also compared two approaches in Figure S7, please see below:

Figure S7. Raw score versus z-score for asymmetry index.

In addition, converting asymmetry to angle using arcsin or arctan doesn't change the data variation between regions, so we did not pursue this further.

Conversely, we used (LH-RH)/(LH+RH) for the functional connectivity matrix. Here all values are positive, and the connectivity matrices are Fisher Z-transformed and we account for any possible local variability in intensity to get at a relative measure of asymmetry. However, results

were similar irrespective of whether we used (LH-RH)/(LH+RH) or LH-RH to compute functional asymmetry. Please find the comparison figure below:

Supplementary Figure S9. Comparisons between original asymmetry (LH-RH)/(LH+RH) calculation and LH-RH for functional data.

Q5: “Colour coding in Fig 1 is really challenging to understand over the whole figure.”

A5: We have deleted massive colors and use text instead to indicate the labels now as below.

Q6: “it is unclear meaning in Fig 1C: mean intensity map and residuals. What exactly this figure does?”

A6: We are happy to further clarify. The BigBrain data is a 60*360 (intracortical depth*region) matrix. It is recommended to correct the mean intensity score along the layers (intracortical depth) as a preprocessing step (Paquola, Wael, et al., 2019). To account for the bilateral bias (as a global measure) we z-scored left and right hemispheres separately for in vivo (1*180) data. For BigBrain, we also need to consider the column-wise correction, so the brain map shows residual intensity after regressing out the intensity of the mean surface for left and right hemispheres separately.

Q7: “in Fig 2C, for the colour combination of yellow-white, it is almost impossible to see the correlations”

A7: Thanks! We have now adapted this part. We deleted the color and used black numbers and bold to indicate the significance.

Q8: “I did not find any tables in the supplementary.”

A8: Apologies for this omission! Now we offer the source data for every datasets we used with the resubmission, which can be used for potential meta-analysis or enrichment analysis.

Q9: “Often the authors referred to significant correlations with p-values, however, it is more informative to see in parallel the effect size.”

A9: We offered the Pearson r or Cohen’s d as effect size for the correlations and comparisons, as well as provided in source data now.

Q10: “I have read a few times what authors would like to say in Fig. 3 and it was too entangled in both in the text and figure captions. Authors should elaborate it in order to give a more clear message.”

A10: We are happy to clarify. Region-wise coupling is a correlation between microstructural asymmetry at the region level (180 regions) and a (180*180) seed-based functional asymmetry map. Following a map of regional coupling between microstructure and function (180 r values) can be obtained and plotted on the cortical surface. We can compute this both at group and individual levels. The individual co-variation is using the individual differences in both microstructural and functional asymmetry. Here, we have 1004 subjects for both microstructural and functional data. Then we computed the link between microstructural asymmetry and functional asymmetry as follows: one region: (region x , 1 microstructural asymmetry marker) and (region x , 180 functional asymmetry markers of its connectivity), correlate along the “region x ” axis to obtain 180 r values for one region, indicating how this region’s microstructural asymmetry relates to the asymmetry of functional connection to this region. We repeated this analysis for all 180 regions to obtain a 180*180 matrix. Similar to the functional gradient approach in the FC matrix, we calculate the affinity matrix on this and decompose it to obtain the PC vectors and loadings. We now revise it further in the Methods section. See below:

*“Second, we correlated the microstructural asymmetry with the FC asymmetry spatially at the group and individual levels. In particular, the correlation coefficient was computed between microstructural asymmetry (in 180 regions) and the (180*180) seed-based functional asymmetry map resulting in a regional coupling score. Following, a map of regional coupling between microstructure and function (180 r values) was obtained for all 180 functional seeds. Regarding the covariation across subjects, for a given region, we did the following: (region x , one microstructural asymmetry marker) and (region x ,*

*180 functional asymmetry markers of its connectivity), were correlated along the “region x” axis to obtain 180 r values for this region, indicating how this region’s microstructural asymmetry supports asymmetry of functional connection to this region across subjects. This procedure was repeated for all 180 parcels. The 180*180 covariation matrix can be obtained with the columns as microstructural profiles and rows as functional profiles. Then, principal component analysis (PCA) was used to decompose the affinity of the covariance matrix with the sparsity of top 10% scores. These principal components reflect the organization features of asymmetric microstructure-function coupling, e.g., two regions having similar asymmetric coupling profiles get close loadings along the PCs.”*

Q11: “It is much better to find a number for "similarity" every time it is used in the text. In particular, when it concerns spatial patterns including differences in layers.”

A11: We agree, and we have now quantified each similarity evaluation using Pearson's *r*.

Reference

- Amunts, K., Lepage, C., Borgeat, L., Mohlberg, H., Dickscheid, T., Rousseau, M.-É., Bludau, S., Bazin, P.-L., Lewis, L. B., Oros-Peusquens, A.-M., Shah, N. J., Lippert, T., Zilles, K., & Evans, A. C. (2013). BigBrain: An Ultrahigh-Resolution 3D Human Brain Model. *Science*, *340*(6139), 1472–1475. <https://doi.org/10.1126/science.1235381>
- Amunts, K., Mohlberg, H., Bludau, S., & Zilles, K. (2020). Julich-Brain: A 3D probabilistic atlas of the human brain's cytoarchitecture. *Science*, *369*(6506), 988–992. <https://doi.org/10.1126/science.abb4588>
- Burt, J. B., Helmer, M., Shinn, M., Anticevic, A., & Murray, J. D. (2020). Generative modeling of brain maps with spatial autocorrelation. *NeuroImage*, *220*, 117038. <https://doi.org/10.1016/j.neuroimage.2020.117038>
- Glasser, M. F., Coalson, T. S., Harms, M. P., Xu, J., Baum, G. L., Autio, J. A., Auerbach, E. J., Greve, D. N., Yacoub, E., Van Essen, D. C., Bock, N. A., & Hayashi, T. (2022). Empirical transmit field bias correction of T1w/T2w myelin maps. *NeuroImage*, *258*, 119360. <https://doi.org/10.1016/j.neuroimage.2022.119360>
- Glasser, M. F., & Essen, D. C. V. (2011). Mapping Human Cortical Areas In Vivo Based on Myelin Content as Revealed by T1- and T2-Weighted MRI. *Journal of Neuroscience*, *31*(32), 11597–11616. <https://doi.org/10.1523/JNEUROSCI.2180-11.2011>
- Glasser, M. F., Sotiropoulos, S. N., Wilson, J. A., Coalson, T. S., Fischl, B., Andersson, J. L., Xu, J., Jbabdi, S., Webster, M., Polimeni, J. R., Van Essen, D. C., & Jenkinson, M. (2013). The minimal preprocessing pipelines for the Human Connectome Project. *NeuroImage*, *80*, 105–124. <https://doi.org/10.1016/j.neuroimage.2013.04.127>
- Haast, R. A. M., Ivanov, D., Formisano, E., & Uludağ, K. (2016). Reproducibility and Reliability of Quantitative and Weighted T1 and T2* Mapping for Myelin-Based Cortical Parcellation at 7 Tesla. *Frontiers in Neuroanatomy*, *10*, 112. <https://doi.org/10.3389/fnana.2016.00112>
- Kochunov, P., Patel, B., Ganjgahi, H., Donohue, B., Ryan, M., Hong, E. L., Chen, X., Adhikari,

- B., Jahanshad, N., Thompson, P. M., Van't Ent, D., den Braber, A., de Geus, E. J. C., Brouwer, R. M., Boomsma, D. I., Hulshoff Pol, H. E., de Zubicaray, G. I., McMahon, K. L., Martin, N. G., ... Nichols, T. E. (2019). Homogenizing Estimates of Heritability Among SOLAR-Eclipse, OpenMx, APACE, and FPHI Software Packages in Neuroimaging Data. *Frontiers in Neuroinformatics*, *13*, 16. <https://doi.org/10.3389/fninf.2019.00016>
- Marques, J. P., Kober, T., Krueger, G., van der Zwaag, W., Van de Moortele, P.-F., & Gruetter, R. (2010). MP2RAGE, a self bias-field corrected sequence for improved segmentation and T1-mapping at high field. *NeuroImage*, *49*(2), 1271–1281. <https://doi.org/10.1016/j.neuroimage.2009.10.002>
- Paquola, C., Bethlehem, R. A., Seidlitz, J., Wagstyl, K., Romero-Garcia, R., Whitaker, K. J., Vos de Wael, R., Williams, G. B., NSPN Consortium, Vértes, P. E., Margulies, D. S., Bernhardt, B., & Bullmore, E. T. (2019). Shifts in myeloarchitecture characterise adolescent development of cortical gradients. *eLife*, *8*, e50482. <https://doi.org/10.7554/eLife.50482>
- Paquola, C., & Hong, S.-J. (2023). The Potential of Myelin-Sensitive Imaging: Redefining Spatiotemporal Patterns of Myeloarchitecture. *Biological Psychiatry*, *93*(5), 442–454. <https://doi.org/10.1016/j.biopsych.2022.08.031>
- Paquola, C., Wael, R. V. D., Wagstyl, K., Bethlehem, R. A. I., Hong, S.-J., Seidlitz, J., Bullmore, E. T., Evans, A. C., Misic, B., Margulies, D. S., Smallwood, J., & Bernhardt, B. C. (2019). Microstructural and functional gradients are increasingly dissociated in transmodal cortices. *PLOS Biology*, *17*(5), e3000284. <https://doi.org/10.1371/journal.pbio.3000284>
- Quinn, B. P. A., Watson, D. M., Noad, K., & Andrews, T. J. (2024). Idiosyncratic patterns of interhemispheric connectivity in the face and scene networks of the human brain. *Imaging Neuroscience*, *2*, 1–20. https://doi.org/10.1162/imag_a_00181
- Raemaekers, M., Schellekens, W., Petridou, N., & Ramsey, N. F. (2018). Knowing left from right: Asymmetric functional connectivity during resting state. *Brain Structure and Function*,

223(4), 1909–1922. <https://doi.org/10.1007/s00429-017-1604-y>

Vos de Wael, R., Benkarim, O., Paquola, C., Lariviere, S., Royer, J., Tavakol, S., Xu, T., Hong, S.-J., Langs, G., Valk, S., Misic, B., Milham, M., Margulies, D., Smallwood, J., & Bernhardt, B. C. (2020). BrainSpace: A toolbox for the analysis of macroscale gradients in neuroimaging and connectomics datasets. *Communications Biology*, 3(1), 1–10. <https://doi.org/10.1038/s42003-020-0794-7>

Wan, B., Bayrak, Ş., Xu, T., Schaare, H. L., Bethlehem, R. A., Bernhardt, B. C., & Valk, S. L. (2022). Heritability and cross-species comparisons of human cortical functional organization asymmetry. *eLife*, 11, e77215. <https://doi.org/10.7554/eLife.77215>

Weiskopf, N., Suckling, J., Williams, G., Correia, M., Inkster, B., Tait, R., Ooi, C., Bullmore, E., & Lutti, A. (2013). Quantitative multi-parameter mapping of R1, PD*, MT, and R2* at 3T: A multi-center validation. *Frontiers in Neuroscience*, 7. <https://www.frontiersin.org/articles/10.3389/fnins.2013.00095>

Reviewer 1

Q: I think the issue of heritability calculation still remains. The current analysis does not separate genetic influences from shared environmental influences within families. Twin studies are designed to separate these (by applying a standard ACE model rather than an AE model, where C is the shared family environment component). In the rebuttal letter, the authors say that Kochunov et al. (2019) found the AE model to be more accurate for estimating heritability than ACE in the HCP dataset. This finding was not obvious to me in Kochunov's results or recommendations (different analysis methods were applied so that the specific effect of ACE vs AE models may not have been clear). Also the brain measures in the present study are not the same as studied by Kochunov et al.. Regardless of the findings by Kochunov et al., an ACE model would better be tested as a sanity check on the AE results in the present study. Otherwise the reader is left wondering whether there is significant heritability at all, or whether the results reflect a familial effect of shared environment instead.

(By the way, the revised text on heritability in the revised Methods section of the manuscript (page 23) cites reference number 79, but in the reference list 79 is a different paper, and Kochunov et al. does not appear. Please check citation-reference matching.)

Thank you for describing what the skew measure indicates in the revised Results text. I think the Methods section still misses the exact formulation of this measure.

A: We thank the reviewer for the suggestion. We now calculated heritability using the A+C+E model and provided Supplementary Figure S11 and Source Data. We revise the methods section below:

“We also included the A+C+E model in Supplementary Figure S11.”

Figure S11. T1w/T2w asymmetry heritability in HCP (N = 1101): A + E versus A + C + E model.

The reference number has changed to 105. Formulation of skewness has been offered in Methods section, see below:

“Regarding the asymmetry of skewness in BigBrain, the skewness formula was used: skewness = $\text{sum}((\text{intensity}_{\text{surface}} - \text{mean})^3) / \text{SD}^3$, where mean and SD are calculated across the sixty surfaces.”